# Direct effect of aerosols on solar radiation and gross primary production in boreal and hemiboreal forests

Ekaterina Ezhova[1], Ilona Ylivinkka[1], Joel Kuusk[2], Kaupo Komsaare[3], Marko Vana[3], Alisa Krasnova[4], Steffen Noe[4], Mikhail Arshinov[5], Boris Belan[5], Sung-Bin Park[6], Jošt Valentin Lavrič[6], Martin Heimann[1,6], Tuukka Petäjä[1], Timo Vesala[1,7], Ivan Mammarella[1], Pasi Kolari[1], Jaana Bäck[7], Üllar Rannik[1], Veli-Matti Kerminen[1], and Markku Kulmala[1]

[1]Institute for Atmospheric and Earth System Research / Physics, Faculty of Science, University of Helsinki, P.O. Box 64, 00014 Helsinki, Finland
[2]Tartu Observatory, Faculty of Science and Technology, University of Tartu, Tõravere, Nõo parish, 61602 Tartu county, Estonia
[3]Institute of Physics, Faculty of Science and Technology, University of Tartu, 50411, Tartu, Estonia
[4]Department of Plant Physiology, Institute of Agricultural and Environmental Sciences, Estonian University of Life Sciences, EE-51006 Tartu, Estonia
[5]V.E. Zuev Institute of Atmospheric Optics of Siberian Branch of the Russian Academy of Sciences, 634055 Tomsk, Russia
[6]Max Planck Institute for Biogeochemistry, Hans-Knöll-Str. 10, 07745 Jena, Germany
[7]Institute for Atmospheric and Earth System Research / Forest Sciences, Faculty of Science, University of Helsinki, P.O. Box 64, 00014 Helsinki, Finland
*Correspondence to:* Ekaterina Ezhova (ekaterina.ezhova@helsinki.fi)

**Abstract.** The effect of an aerosol loading on solar radiation and further on photosynthesis is a relevant question for estimating climate feedback mechanisms. This effect is quantified in the present study using ground-based measurements from five remote sites in boreal and hemiboreal (coniferous and mixed) forests of Eurasia. The diffuse fraction of global radiation associated with the direct effect of aerosols, that is excluding the effect of clouds, increases with an increasing aerosol loading. The increase in the diffuse fraction of global radiation from approximately 0.11 on the days characterized by low aerosol loading up to 0.2 - 0.27 pertaining to relatively high aerosol loading leads to the increase in gross primary production (GPP) at all sites by 6-14%. The largest increase in GPP (relative to the days with low aerosol loading) is observed for two types of ecosystems: a coniferous forest at the high latitudes and a mixed forest at the middle latitudes. For the former ecosystem the change in GPP due to relatively large increase in the diffuse radiation is compensated by the moderate increase in the light use efficiency. For the latter ecosystem, the increase in diffuse radiation is smaller for the same aerosol loading, but the smaller change in GPP due to this relationship between radiation and aerosol loading is compensated by the higher increase in the light use efficiency. The dependency of GPP on the diffuse fraction of solar radiation has a weakly pronounced maximum related to clouds.

## 1 Introduction

According to the IPCC report (Pachauri et al., 2014), the influence of aerosol loading on solar irradiance remains a relevant open question. Aerosol particles influence the radiative balance of the Earth by aerosol-radiation and aerosol-cloud interactions.

The aerosol-radiation interaction includes scattering and absorption of solar radiation (direct effect), and potential changes in cloud properties associated with these radiative effects (semi-direct effect). Another important consequence of aerosol-radiation interactions is an increase in the diffuse fraction of solar radiation entering the Earth surface.

The influence of aerosol-radiation interaction on the diffuse fraction of solar radiation is relevant for the estimates of the terrestrial carbon sink and in understanding of climate feedback mechanisms (e.g., Kulmala et al., 2014). The increase in the land carbon sink due to the enhanced industrial aerosols was estimated to be 20-25% during the period of 'global dimming' between 1960 and 1999 in the modelling study by Mercado et al. (2009). The physical mechanism behind the growth of the terrestrial carbon sink is as follows. An increased diffuse fraction of solar irradiance due to aerosol makes it easier for light photons to penetrate into the canopy. Moreover, diffuse light coming from different angles increases the efficiency of $CO_2$ uptake by leaves of different orientation (Alton et al., 2007). This leads to an increase in the light use efficiency of plants (LUE, quantifying the amount of $CO_2$ fixed by an ecosystem per unit photosynthetically active radiation) and gross primary production (GPP, quantifying the amount of $CO_2$ fixed by an ecosystem per unit area per unit time). This mechanism, however, is ecosystem-dependent and presumably depends on canopy height and leaf area index (Niyogi et al., 2004; Kanniah et al., 2012; Cheng et al., 2015): an enhanced diffuse radiation does not lead to an increase in grasslands' GPP, while GPP of broadleaf forests can be significantly increased. Similarly, the study based on AmeriFlux data from several sites (Cheng et al., 2015), including broadleaf forests and crops, suggests an increase in GPP due to diffuse radiation for forests rather than for crops.

Estimates of aerosol effect on GPP are uncertain for two reasons. First, it is not clear how large aerosol effect on diffuse radiation is. Second, the associated effect of diffuse radiation on GPP can be both negative and positive (Niyogi et al., 2004; Alton, 2008; Park et al., 2018). Therefore, aerosol-radiation interaction may lead either to increase or decrease in GPP, depending on aerosol loading. As an example, Niyogi et al. (2004) reports an increase in GPP for broadleaf forests at any aerosol loading typical for the sites they considered. On the contrary, a recent model study suggests that for a substantial part of boreal forests on the territories of Finland, Estonia and Russia, the direct aerosol effect at relatively high aerosol loading leads to a decrease in the annual diffuse irradiance and GPP (Lu et al., 2017). Often estimates on the effects of solar radiation on GPP have been obtained involving parametrizations (Alton, 2008), based on the results of numerical modelling and satellite observations (Lu et al., 2017). The aims of this study are to provide a comprehensive data analysis related to the direct effect of aerosol on solar radiation, to separate cloud and aerosol effects on solar irradiance, and to further quantify the influence of solar radiation on GPP. The data sets include continuous field measurements from five stations in Finland, Estonia and Russia.

Note that this analysis can be put into a broader context regarding the quantification of terrestrial feedback loops, e.g. the COntinental Biosphere-Atmosphere-Cloud-Climate (COBACC) feedback loop (Kulmala et al., 2014). This loop considers the effect of carbon dioxide concentration on biogenic volatile organic compound emissions (BVOC), further relates BVOC-aerosol interactions to the variability in solar radiation and finally the loop is closed with the effects of radiation for the ecosystem GPP and $CO_2$ concentration. Here we provide the quantification of the part of this loop related to aerosol - solar radiation - photosynthesis interactions in boreal and hemiboreal forests.

In line with the aims of this study, we focus on two problems: aerosol-radiation interaction, for which we quantify the diffuse fraction of solar radiation that can be observed due to direct aerosol effect, and further the diffuse radiation effect

on photosynthesis. First, we quantify the effect of aerosol-radiation interaction on diffuse radiation in boreal forests. For the estimates of the effect of aerosol on solar radiation it is important to separate clear time of the day from cloudy time, because clouds are much more effective than aerosols in scattering of solar irradiance. In some of previous studies (Gu et al., 2002; Kulmala et al., 2014) this separation was made based on the ratio between actually measured global irradiance and that theoretically calculated at the top of the atmosphere. This criterion of clear days can indeed be acceptable when one needs to distinguish between mostly sunny/mostly cloudy days from the point of view of the incoming global irradiance or the incoming solar energy. However, it fails to separate sunny/cloudy time from the point of view of diffuse irradiance amount, which is crucially important for photosynthesis. Generally, radiation data used for analyses is averaged over some time interval (often half an hour). Some types of clouds, such as cumulus, are rather intermittent with a period of oscillations of several minutes (e.g., Duchon and O'Malley, 1999). During the periods when there is no cloud in front of the sun, global irradiance can be even higher than theoretically predicted due to a global radiation enhancement (Pecenak et al., 2016). As a result, an averaged global irradiance can be close to that theoretically predicted for the clear sky, while at the same time an averaged diffuse irradiance is significantly enhanced due to the presence of clouds. Such data can be erroneously attributed to clear sky conditions, so that the effects of clouds on the diffuse radiation can be associated with the direct aerosol effect. These kind of conditions are most likely to cause the highest GPP, because global irradiance does not decrease while diffuse irradiance is already high. Second, we consider the effect of solar radiation on GPP. We investigate the effect of diffuse fraction of solar radiation on LUE and further quantify the maximum effect on GPP due to aerosol-radiation interaction for different ecosystems in boreal forest.

## 2   Materials and methods

In this section we introduce data sets and methods used in the study. In subsection 2.1 we present the sites and data sets. Subsection 2.2 describes the clear sky model used to separate clear sky and clouds. We discuss the consequences of having different (PAR or broadband) radiation measurements at different sites in Subsection 2.3 and suggest a method to make the data sets comparable. In Subsection 2.4 we introduce condensation sink as a measure of aerosol loading and compare it to aerosol optical depth at 500 nm, typically used to quantify aerosol loading in model and satellite-based studies. Finally, subsection 2.5 describes the method to study diffuse radiation effect on ecosystem GPP using a separation of GPP into LUE and PAR.

### 2.1   Sites and data sets

We used data from five remote forest sites located at the middle and relatively high latitudes in Finland, Estonia and Russia: SMEAR I (Värriö, Finland), SMEAR II (Hyytiälä, Finland), SMEAR Estonia (Järvselja, Estonia), Zotino (Zotino, Krasnoyarsk region, Russia) and Fonovaya (Tomsk region, Russia). Fig. 1 illustrates sites' location on the map, while Table 1 summarizes the information on the data sets used in the present study. The information on instruments can be found from the separate papers describing the stations (listed below) and AVAA Internet site for SMEAR I and II (*https://avaa.tdata.fi/web/smart*).

SMEAR II (Hari and Kulmala, 2005) is located at Hyytiälä Forestry Field Station in a conifer boreal forest near the lake, central Finland. The site is a managed, 55-year old Scots pine (*Pinus sylvestris L.*) forest stand with closed canopy and average tree height ca 18 m.

SMEAR I (Hari et al., 1994) is the site characterized by the highest latitude, it is located in Finnish Lapland. The site has ca 60-year old Scots pines with a rather open canopy, average tree height is ca 10 m.

SMEAR Estonia (Noe et al., 2015) is located in a hemiboreal forest zone and the stands in the tower footprint consist of a mixture of coniferous (Scots pine and Norway spruce (*Picea abies L. Karst*)) and deciduous (silver birch (*Betula pendula Roth.*) and downy birch (*Betula pubescens Ehrh.*)) trees. Because of the high heterogeneity and the mosaic of stands of hemiboreal forests the stand age is on average 65 years ranging from 43-year old larch stands to 120-year old pine stands. The average age of the dominating species are 102 years for pine, 79 years for spruce and 68 years for birch. Also tree height is therefore variable and it is 22 m on average (ranging between 10 m and 30 m).

Fonovaya, Tomsk region is the most southern site (Matvienko et al., 2015). It is located on the river Ob' in West Siberia, Russia, and forest stand is represented by mixed forest. The stand is dominated by 50-year old Scots pine, 45-year old birch (*Betula verrucósa*) and 27-year old aspen (*Populus tremula*). Average tree height is ca 30 m, ranging from 25 m for birch and up to 40 m for pines.

Zotino (Heimann et al., 2014; Park et al., 2018) is located near the river Yenisei in Central Siberia. The forest is dominated by 100-year old Scots pines (*Pinus sylvestris L.*) forest stand with open canopy and average canopy height ca 20 m.

Thus, we have data sets from five sites with three of them represented by pine stands and two by mixed forests, all of them located at the middle latitudes. Note that currently the data from these five stations form the largest possible set of simultaneous atmospheric observations on trace gases, meteorology, solar radiation and aerosols, conducted in boreal and hemiboreal forests in Eurasia. Some of these sites lack certain components necessary for the analysis and therefore we use parametrizations when needed (and possible). For example, the diffuse fraction of solar radiation was parameterized in the dataset from Fonovaya. Following Gu et al. (2002) and Alton (2008), we used the formula

$$fdif_{bb} = R_d/R_g = 1.45 - 1.81x, \tag{1}$$

where $x = R_g/R_{TOA}$ is the ratio of the measured global irradiance to the modelled irradiance on the top of atmosphere. For $x < 0.28$ we used $fdif_{bb} = 0.95$, and for $x > 0.75$ we used $fdif_{bb} = 0.1$. The radiation on the top of atmosphere was calculated as $R_{TOA} = I_0\cos(sza)$, where $I_0$ is the solar constant and $sza$ is the solar zenith angle.

The data used in this study correspond to the peak growing season defined as the time period with maximum GPP. Typically it includes June-August and partly May and September for all the sites except SMEAR I, where it includes July-August and part of June and September. For consistency, we used June-August data for SMEAR II, SMEAR Estonia, Fonovaya and Zotino and July-August data for SMEAR I. The analysis is performed for daytime (09:00 – 15:00 local time), half-an-hour averaged data.

Aerosol number-size distribution at Fonovaya was measured with two instruments, which do not overlap in the size range. Therefore, there is a gap between 200 nm and 300 nm in the distribution. This gap was filled with an average value between the

two adjacent points (one point was added between these two points). Apart from that, we did not apply gap-filling for aerosol data in this study, using only good quality data. We did not fill gaps in solar radiation data.

Note, that $CO_2$ flux was obtained by the micrometeorological (eddy covariance) method at SMEAR stations and Zotino, while the gradient method was applied to obtain $CO_2$ flux from Fonovaya data set. GPP was calculated from the formula

$$\text{GPP = TER - NEE,} \tag{2}$$

where TER is the total ecosystem respiration and NEE is the net ecosystem exchange. TER was obtained for all ecosystems using the nighttime method of $CO_2$ flux partitioning (Reichstein et al., 2005). The partitioning method at SMEAR I and II was based on soil temperature which makes the method more precise (Kolari et al., 2009; Lasslop et al., 2012), while at other sites it was based on air temperature (soil temperature is not measured). According to Lasslop et al. (2012), a possible consequence for average GPP calculated during daytime (9:00-15:00), is a decrease in GPP calculated using soil temperature by about 0.5 $\mu$mol s$^{-1}$ m$^{-2}$ as compared to GPP calculated using air temperature. In addition, in the data sets from SMEAR stations absent GPP points were modelled (Kolari et al., 2009), while in the data sets from the Russian stations we did not fill the gaps. For Zotino set good data percentage was on average 55% (Park et al., 2018), and for Fonovaya set it was approximately 30%.

Studying aerosol-radiation interactions, we used clear sky model Solis (see more in Sec. 2.2). The input parameters for this model are aerosol optical depth at 700 nm ($AOD_{700}$) and precipitable water (PW). We used AOD at 675 nm and PW from Aeronet (Holben et al., 1998); in particular, from the following Aeronet sites: Hyytiala (for SMEAR II), Sodankyla (for SMEAR I), Tomsk22 (for Fonovaya) and Toravere (for SMEAR Estonia). Aeronet sites are in immediate vicinity of Fonovaya and SMEAR II stations, SMEAR Estonia is 50 km away from Toravere and SMEAR I is approximately 70 km away from Sodankyla. We used Version 2, Level 2 data (cloud screened and quality controlled), except for Tomsk22 where we used Version 3, Level 2 data. The input data were averaged over daytime.

Currently aerosol characteristics are not measured at Zotino and there are no Aeronet sites nearby, therefore aerosol-radiation interaction has not been studied for Zotino. However, radiation and $CO_2$ flux, necessary for the radiation-photosynthesis analysis, are measured and therefore Zotino set has been included in this study.

## 2.2 Clear sky model Solis

To distinguish between the effects of aerosol and clouds on solar radiation, we used a simplified broadband version of a clear sky radiative transfer model (RTM) - Solis (Ineichen, 2008). This is a quasi-physical model which means that it employs Lambert-Beer relations for the general estimates of irradiance while using parametrizations for the total optical depths. The input parameters are the aerosol optical depth at 700 nm ($AOD_{700}$) and precipitable water (PW). Parameterizations are developed for the following range of parameters: sea level < altitude < 7000 m, 0 < $AOD_{700}$ < 0.45 and 0.2 cm < PW < 10 cm.

Solis parameterizations were obtained with 'urban' type of aerosol size distribution in the full RTM (Ineichen, 2008). The difference between calculations for 'urban' and 'rural' types of aerosol for the same $AOD_{700} = 0.18$ was reported by Mueller et al. (2004). This value of $AOD_{700}$ is larger than the typical values for all the places that we consider in this study, hence, we expect smaller errors due to the inconsistent aerosol type. The difference reported by Mueller et al. (2004) was negligible for

direct irradiance, whereas global irradiance for 'rural' aerosol was around 20 W/m² larger during the daytime. Given that for clear sky conditions between 9:00 and 15:00 and during the growing season, global irradiance falls down to ∼ 600 W/m², this difference introduces a maximum error of 3%. More tests for several sites in the USA and comparison between Solis and two more simplified clear-sky models, Bird and REST2, were reported by Sengupta and Gotseff (2013). Solis is the optimal model from the point of view of the required parameters. By performing only slightly worse as compared with the two other models, it requires only two input parameters.

We used Solis to model both global and diffuse irradiance. The horizontal global irradiance, $I_{\mathrm{gh}}$, was calculated as follows

$$I_{\mathrm{gh}} = R_{\mathrm{g,mod}} = I_0' \exp\left(-\frac{\tau_{\mathrm{g}}}{\cos^{\mathrm{g}}(sza)}\right) \cos(sza), \tag{3}$$

while the direct irradiance, $I_{\mathrm{dir}}$, was calculated as

$$I_{\mathrm{dir}} = I_0' \exp\left(-\frac{\tau_{\mathrm{b}}}{\cos^{\mathrm{b}}(sza)}\right). \tag{4}$$

Here $I_0'$ is a common modified (enhanced) irradiance defined in (Ineichen, 2008), $\tau_{\mathrm{b}}$ and $\tau_{\mathrm{g}}$ are the direct and global total optical depths, $b$ and $g$ are the fitting parameters, $sza$ is the solar zenith angle. Diffuse radiation can be found from the global radiation balance as

$$I_{\mathrm{dh}} = R_{\mathrm{d,mod}} = I_{\mathrm{gh}} - I_{\mathrm{dir}} \cos(sza). \tag{5}$$

All the parametrizations used in the model are given in (Ineichen, 2008). The algorithm used for calculations is written in Fortran. For the calculation of $sza$, the on-line calculator Solar Position Algorithm (SPA) was used (available from *http://www.nrel.gov*).

### 2.3 Accounting for the difference between PAR and broadband radiation

Note that at different sites the measurement methods of solar radiation are different. At SMEAR II before 2009 and Fonovaya only broadband radiation is measured, while at SMEAR I and Zotino only photosynthetically active radiation (PAR) is measured. At SMEAR II after 2009 both global PAR and broadband global radiation, as well as diffuse PAR are measured. Shortwave broadband radiation, referred to as broadband radiation, is the radiation in the spectral range between 0.3 $\mu$m and 4.8 $\mu$m, while PAR is the radiation relevant for photosynthesis, i.e. in the range of wavelengths between 400 and 700 nm. The former is typically measured with thermopile pyranometers (energy sensors) and reported in energy units [W m$^{-2}$], while the latter is measured with quantum sensors and is reported in quantum units [ $\mu$mol s$^{-1}$ m$^{-2}$].In what follows, we quantify the ratio between global PAR and global broadband radiation, and the ratio between diffuse fraction of PAR and diffuse fraction of broadband radiation.

Following Ross and Sulev (2000), the ratio of PAR [$\mu$mol s$^{-1}$ m$^{-2}$] to the broadband radiation [W m$^{-2}$] is called PAR quantum efficiency $\chi$. By dividing global PAR by broadband global radiation at SMEAR II and finding its mean value (the values were obtained for growing season and years listed in Table 1), $\chi_{\mathrm{glob}} = 2.06$ $\mu$mol s$^{-1}$ W$^{-1}$ was obtained, somewhat higher than 1.81 $\mu$mol s$^{-1}$ W$^{-1}$ reported by Ross and Sulev (2000) for Estonia.

We explain the potential difference between diffuse fraction of PAR and diffuse fraction of broadband radiation as follows. Aerosol particles influence a certain part of the spectra of solar irradiance depending on the particle size distribution (e.g., Seinfeld and Pandis, 2016). This effect can be better understood by using a dimensionless size parameter $\pi d_p/\lambda$, where $\lambda$ is the wavelength of incident irradiance and $d_p$ is the particle diameter. If $\pi d_p/\lambda << 1$, then Rayleigh scattering is prevailing, while $\pi d_p/\lambda >> 1$ means that the scattering properties of the particles are determined by the geometrical optics, so-called geometric scattering. The characteristics of aerosol distribution become important for solar irradiance if $\pi d_p/\lambda \sim 1$. For boreal forests with the particle size distribution governed typically by the well-pronounced mode with the geometric mean diameter $d_p \approx 100$ nm (Dal Maso et al., 2008), the effective interaction of aerosols with solar radiation occurs in the ultraviolet range of the wavelengths (100-400 nm) and in the blue part of the optical spectrum (400-500 nm). As compared to PAR (400-700 nm), the essential part of the broadband radiation energy is contained in the near infrared part of the spectrum (700-1400 nm), which is not affected by the relatively small particles prevailing in aerosol distributions typical for boreal forest. In other words, the effect of aerosols on the diffuse fraction of solar irradiance can be different for PAR and broadband radiation. Qualitatively, one would expect that both the increase in diffuse radiation and the decrease in global radiation would be more pronounced for PAR, so that the diffuse fraction of PAR would be higher under the same aerosol conditions. In order to make our analysis consistent and to be able to compare results from different sites, we performed analysis of the data from SMEAR Estonia to compare diffuse fractions of PAR and broadband radiation. The data set includes four years, from 2014 to 2017.

The measurements at SMEAR Estonia are made using an energy sensor. The hyperspectral radiometer SkySpec (Kuusk and Kuusk, 2018) is a purpose-built instrument for automated measurement of global and diffuse sky irradiance. To obtain PAR radiation, the spectral data are converted from energy units to quantum units and integrated over 400 nm to 700 nm spectral range. Integration over the whole available spectral range in energy units is used for simulating a thermopile pyranometer.

## 2.4 Condensation sink as a measure of aerosol loading

As was mentioned in the Introduction, this study can be considered as a part of the project quantifying COBACC feedback loop using ground-based measurements. Condensation sink (CS) is a typical parameter calculated from a ground-based aerosol number-size distribution and characterizing aerosol loading. It can be related to measured organic vapours, making it possible to study the effect of formation and growth of secondary aerosol for photosynthesis. Besides, CS was chosen in the original study of COBACC feedback loop by Kulmala et al. (2014), therefore, for comparison we resort to this parameter.

CS is calculated from the particle number-size distribution as (Kulmala et al., 2001):

$$\text{CS} = 2\pi D_v \int_0^{d_{p,\max}} d_p \beta n(d_p) dd_p, \tag{6}$$

where $D_v$ is the diffusion coefficient of the condensing vapour, $n(d_p)$ is the particle number-size distribution, and $\beta$ is the Fuchs-Sutugin coefficient. The physical meaning of the CS is the inverse time needed for vapours to be uptaken by existing aerosol particles. Similar to scattering coefficient and AOD, the value of CS depends on an aerosol surface area. However, the contribution of large particles to the CS diminishes with the increasing particles diameter, opposite to an aerosol surface area.

This effect becomes pronounced for the particles with the diameter larger than about 300 nm. For boreal forests, where the mode with a geometric mean diameter of around 100 nm dominates in the particle number-size distribution, CS can be assumed to be directly proportional to the aerosol surface area (Ezhova et al., 2018). Thus, one can expect that CS is an appropriate measure of atmospheric aerosols for the radiation studies in boreal forests. Connection between this parameter and $AOD_{500}$
for boreal forests is discussed in more detail in Appendix A.

## 2.5 LUE and PAR analysis to assess the diffuse radiation effect on GPP

There is a strong evidence that GPP dependence on $R_d/R_g$ is not linear and has a maximum (Alton, 2008; Moffat et al., 2010; Park et al., 2018), but this maximum is not always well pronounced. In what follows we explain the GPP maximum based on the ecosystem LUE and PAR dependences on $R_d/R_g$. Following Cheng et al. (2016), we defined LUE as GPP per unit PAR,
therefore GPP = LUE · PAR. Strictly speaking, LUE is defined as GPP per unit absorbed PAR, i.e. $PAR_{abs} = fAPAR \cdot PAR$, where fAPAR is the fraction of absorbed PAR. The fraction of absorbed PAR depends on leaf area index (LAI), solar zenith angle and other factors. This dependence for boreal forests was studied in (Maijasalmi, 2015; Hovi et al., 2016). Based on results reported by Hovi et al. (2016), fAPAR for tree height larger than 10 m and at a moderate zenith angle (40°-60°) can be estimated as 0.8-0.9. One can obtain LUE defined with absorbed PAR dividing LUE used in the present study by 0.8-0.9.

For all ecosystems with sufficiently deep canopy and high leaf area index, LUE is expected to increase with $R_d/R_g$, as larger fraction of available photons can penetrate inside the canopy and they can be used for photosynthesis. Some studies (e.g., Niyogi et al., 2004) reported a linearly growing dependence of LUE on $R_d/R_g$. Furthermore, a decrease of PAR with $R_d/R_g$ can be expected, because an increase in the diffuse fraction of global irradiance corresponds to the enhancement of the scattering and reflecting properties of the atmosphere due to the presence of aerosols or clouds. Therefore, for each site the
dependences of LUE and PAR on $R_d/R_g$ were investigated separately, after which the GPP dependence on $R_d/R_g$ was derived from these two dependences.

Again, in order to have consistent data sets, we recalculated $R_d/R_g$ obtained from PAR measurements to have it in terms of broadband radiation at all the sites. Conversely, broadband global radiation from Fonovaya and part of SMEAR II data sets was recalculated to PAR when investigating LUE and PAR dependence on $R_d/R_g$. We multiplied global radiation by $\chi_{glob} = 2.06$
$\mu$mol s$^{-1}$ W$^{-1}$ in order to get PAR in quantum units. The PAR quantum efficiency was chosen equal to the one at SMEAR II, since for the daytime and similar solar zenith angles it is mostly aerosol dependent, and SMEAR II and Fonovaya have in general similar aerosol loading which can be confirmed by their similar values of CS (Fig. 5). Considering LUE, we filter out the data having a low global irradiance, $R_g \leqslant 100$ W m$^{-2}$ (PAR $\leqslant 200$ $\mu$mol s$^{-1}$ m$^{-2}$). Below this critical $R_g$, LUE shows significant scatter (being high for the low radiation values); therefore we excluded these data from analysis.

## 3 Results and discussion

We present the results of our study in two subsections. In Subsection 3.1 we report the results related to aerosol effect on solar radiation, and in Subsection 3.2 we report the results related to the effect of diffuse radiation for ecosystem photosynthesis. The link between these results and their relation to other studies are discussed in subsection 3.3.

### 3.1 Aerosol effect on solar radiation

#### 3.1.1 Criterion of clear sky based on clear sky model Solis

To understand the importance of the clear-sky criterion for the diffuse fraction of global radiation, we report the model test against diffuse and global irradiance measurements at SMEAR Estonia (clear sky model Solis is described in Section 2.2). Examples of the global and diffuse radiation diurnal cycles for clear and cloudy days are displayed in Fig. 2. Note that on cloudy day with patchy clouds, AOD can still be measured. Model results for cloudy day report the global and diffuse clear-sky radiation for AOD and PW measured on that day. In general, the model performs well during clear sky conditions as can be seen in the left panel of Fig. 2. This is in accordance with the results of Sengupta and Gotseff (2013) reporting good performance of the model for clear sky conditions at several sites in the US. On a cloudy day (right panel, Fig. 2), there were times (e.g. at 09:00 and 17:00) when the measured and modelled global irradiance ($R_{g,meas}$ and $R_{g,mod}$) were nearly equal while the measured diffuse irradiance ($R_{d,meas}$) was significantly higher than the modelled one ($R_{d,mod}$). The criterion of clear sky based on the comparison between the modelled and measured global irradiance, i.e. involving only global irradiance (e.g., Kulmala et al., 2014), does not filter out these points.

A further illustration of the simplified criterion and its consequences for the diffuse fraction of global irradiance is given in Fig. 3, displaying the diffuse fraction of global irradiance under clear sky conditions at SMEAR Estonia (summer 2016). The open symbols together with the closed symbols correspond to the data obtained using the clear sky criterion involving only global radiation:

$$R_{g,meas}/R_{g,mod} \geq 0.9 \tag{7}$$

while the closed symbols show the data obtained with the criterion involving both diffuse and global radiation:

$$R_{g,meas}/R_{g,mod} \geq 0.9 \text{ and } 0.8 \leq R_{d,meas}/R_{d,mod} \leq 1.2. \tag{8}$$

The criterion of clear sky suggested here is based on the results of Sengupta and Gotseff (2013), who determined an r.m.s. error of the linear regression corresponding to the measured global radiation and modelled one using Solis for eight sites. This r.m.s. error did not exceed 10% (cf. criterion (7) using global radiation), and generally was lower than that. As for diffuse radiation, we took 20% difference between measured and modelled diffuse radiation in criterion (8). This difference was chosen based on the estimated 18 W m$^{-2}$ error in diffuse radiation between the full radiative transfer model Solis and measurements, reported by Ineichen (2008), and by assuming a typical diffuse radiation value of 120 W m$^{-2}$, and by adding 5% error between the simplified model and the full radiative transfer model Solis. Closed symbols in Fig. 3 show a good agreement between

the measured and modelled diffuse fractions of solar irradiance $R_d/R_g$, while a considerable part of open symbols has large measured $R_d/R_g$, that is, they contain a big fraction of cloud-influenced data. Therefore, the criterion of clear sky based on the global and diffuse radiation can be used to detect clear sky data when it is important to separate the effect of aerosol and clouds on diffuse radiation.

Note that the diffuse fraction of solar irradiance due to aerosol loading at SMEAR Estonia lies between 0.08 and 0.21. As shown later, this relatively low ratio pertains to all the sites under this study: the maximum diffuse fraction of solar irradiance due to the direct aerosol effect was no more than 0.27 at the remote sites in boreal forests during the growing season.

### 3.1.2  Parameterization of the diffuse fraction of PAR as a function of the diffuse fraction of broadband radiation

In this section we discuss the difference between the diffuse fractions of PAR and broadband radiation. Fig. 4 displays the ratio
between the diffuse fraction of PAR, $fdif_{PAR}$, and the diffuse fraction of broadband radiation, $fdif_{bb}$, as a function of $fdif_{bb}$ where $fdif = R_d/R_g$. Since the radiation level depends on a solar zenith angle, we cast the daytime data into three solar zenith angle bins, the width of each bin being 10°. Each data set was fitted by the exponential function

$$\frac{fdif_{PAR}}{fdif_{bb}} = a\exp(-(fdif_{bb} - b)/c) + d. \tag{9}$$

The coefficients of the fitting function for each bin are reported in Table 2. Using this function, we can compare the diffuse
fractions of PAR and broadband radiation over the whole range of sky conditions, including clear and cloudy skies. As expected (see more in Section 2.3), we observed in Fig. 4 an increase in the diffuse fraction of PAR up to 27% as compared to the value for broadband radiation at small $fdif_{bb}$, corresponding to clear-sky conditions. In absolute values, this difference between the diffuse fraction of the PAR and broadband radiation is not very large (e.g., $fdif_{bb} = 0.15$ corresponds to $fdif_{PAR} = 0.18$ under clear sky). However, since the diffuse fraction of global radiation in boreal forests varies in a relatively small range due
to the direct effect of aerosol (e.g., Fig.3), it is important to make corrections. As can be further noted from Fig. 4, the ratio $fdif_{PAR}/fdif_{bb}$ approaches one for overcast cloudy conditions, as in this case diffuse radiation prevails for both PAR and broadband radiation.

We use these results to obtain the diffuse fraction of global broadband radiation for the sites where only PAR was measured. In what follows we use the term 'diffuse fraction of global radiation' for broadband radiation.

### 3.1.3  Aerosol influence on the diffuse fraction of global irradiance: comparative analysis for four sites

In this section, we consider the effect of aerosol on the diffuse fraction of global irradiance. In the following analysis the data were filtered to include only clear sky conditions, based on the modelled and measured global irradiance, using criterion (7) for all the sites. We deliberately used the criterion based only on the global irradiance in order to demonstrate the effect of unfiltered cloud-contaminated data on the diffuse fraction of solar radiation. Fig. 5 displays $R_d/R_g$ vs CS at SMEAR I and
II, SMEAR Estonia and Fonovaya (no aerosol data are available from Zotino). To separate the effect of clouds and aerosol particles, we report two quantities: the measured diffuse fraction of global irradiance (blue symbols) and the modelled diffuse fraction of global irradiance (orange symbols). Based on the analysis in Subsection 3.1, modelling provides information about

the direct effect of aerosol on the diffuse fraction, while measurements illustrate the combined effects due to aerosols and clouds. Note that for consistency all the ratios $R_d/R_g$ were corrected in accordance with the previous section, that is, only the ratios corresponding to the broadband radiation are reported (though PAR is measured at SMEAR I and II). For SMEAR I, the model data set includes four years, while only two years of measured data are available. Diffuse radiation is not measured at Fonovaya station, hence only model results are shown. Moreover, the data from 2016 were not used due to the forest fires in Siberia. Smoke plumes have large influences on the aerosol size distribution as a result of which the clear-sky model fails to predict diffuse and global radiation.

An increase in $R_d/R_g$ with increasing CS is observed at all the sites, as follows from the model results (also representative for the measurements with an appropriate clear-sky criterion, as discussed in section 3.1.1 and demonstrated in Fig. 4). Note that the modelled values of $R_d/R_g$ correspond to the lower points in the measured data sets. The blue points above the modelled data are characterized by a larger diffuse radiation than those obtained for current AODs using Solis, and hence represent the effect of clouds. According to the model calculations, the maximum diffuse fraction of global radiation due to the direct effect of aerosol did not exceed 0.27, while the minimum fraction was about 0.1. Also, aerosol population with CS smaller than approximately 0.005 s$^{-1}$ do not contribute significantly to light scattering (Fig. 5), since the diffuse fraction of global irradiance for these values of CS is almost constant and close to 0.1, which can be attributed mostly to Rayleigh scattering.

We fitted modelled and measured data with the linear function $fdif_{bb} = kCS + b$. The best-fit coefficients and correlation coefficients for four sites are reported in Table 3. All the dependences pertaining to modelled data, i.e., to the direct effect of aerosol particles on solar radiation, had the correlation coefficients larger than 0.5 corresponding to moderate correlation (except Fonovaya, where R = 0.44 supposedly because of the small data set). On the contrary, cloud-influenced data demonstrate rather weak correlations with 0.18 < R < 0.33.

## 3.2 The effect of diffuse radiation on gross primary production

### 3.2.1 Diffuse radiation effect on GPP: comparative analysis for all the sites

In this section we study LUE and PAR dependences on the diffuse fraction of global radiation in order to better understand the behaviour of GPP dependence on $R_d/R_g$. Fig. 6 displays the dependences of LUE on $R_d/R_g$ for all sites. All these dependences exhibit a linear relationship with the correlation coefficients between R = 0.67 and R = 0.83 (except Fonovaya with R = 0.44, which can be attributed to both short data set and less precise gradient method used for the CO$_2$ flux calculations). The LUE slope reflects the canopy properties, i.e. it characterizes the ability of a forest stand to uptake more CO$_2$ in response to an increasing diffuse fraction of solar irradiance. The steepest LUE slopes pertain to the mixed forests at SMEAR Estonia and Fonovaya, while the slopes are approximately 60% less steep in coniferous forests (Table 4). This difference is presumably due to the forest type, as mixed forests have a larger potential for the photosynthetic activity enhancement due to a larger leaf area index and deeper canopy. We emphasize that the difference is seen in LUE, in accordance with LUE definition given in section 2.5, which includes the dependence on LAI and tree height attributed to fAPAR in the standard definition. Note that

the increase in LUE from approximately 0.01 to 0.03-0.04 mol $CO_2$ mol photons$^{-1}$ observed for mixed forests is similar to that reported by Cheng et al. (2016) for mixed and broadleaf forests in the United States of America.

Fig. 7 displays the dependences of the global PAR on $R_d/R_g$ for all sites. As expected, PAR decreases with an increasing $R_d/R_g$: at smaller values due to aerosol particles, and at larger values due to clouds. As follows from Fig. 5, the values of $R_d/R_g < (0.2 - 0.27)$ correspond mostly to the influence of aerosol particles (but they can also be influenced by thin clouds), while larger values of $R_d/R_g$ are associated with the presence of clouds. Similarly to LUE, these dependences are linear with high correlation coefficients (0.78 < R < 0.90). Generally, the slopes of the linear dependences in Fig. 7 were similar (within the range 1081-1194 $\mu$mol s$^{-1}$ m$^{-2}$), which can probably be attributed to similar cloud attenuating properties over all the sites at the middle latitudes. The exception is SMEAR I, where the slope is lower (944 $\mu$mol s$^{-1}$ m$^{-2}$). Solar radiation under clear sky conditions is also significantly lower at SMEAR I as compared to other places, which is partly due to the high latitude, and partly because the growing season at SMEAR I is July and August (i.e., it does not include June with the highest global irradiance values).

The vertical scattering of the data in Fig. 7 is presumably due to two factors: first, due to the variability in the radiation intensity during daytime and growing season, and second, due to the different influences of clouds, as the same diffuse fraction of global irradiance may pertain to the different attenuations of global radiation by clouds. Note that the latter factor is excluded from the Fonovaya data set by the parameterization. One can conclude that the PAR variability due to clouds was larger than the diurnal (associated with different solar zenith angles during the day) and day-to-day PAR variability in the growing season. Thus, additional binning by, e.g. solar zenith angle, would be redundant, since the decrease in PAR variability due to binning would be hidden by the stronger scattering due to clouds.

Finally, based on the linear dependences of LUE on $R_d/R_g$ and PAR on $R_d/R_g$, we can estimate how GPP depends on $R_d/R_g$. When we multiplied LUE by PAR, a parabolic dependences were obtained for all the sites, with a maximum due to the effect of diffuse radiation on photosynthesis. Fig. 8 shows estimated GPP dependences on $R_d/R_g$ for different sites for comparison, while Fig.9 displays data sets together with estimated curves separately for all sites, similar to Figs. 6 and 7.

### 3.2.2 Constraints on LUE and diffuse fraction of solar radiation associated with the maximum ecosystem GPP under diffuse light

Well-pronounced linear dependences of LUE and PAR on $R_d/R_g$ can be used to estimate how large an increase in LUE should be in order to have GPP increase under diffuse radiation and at what diffuse fraction of solar radiation the maximum GPP can be observed. If

$$\text{LUE} = L_1 + L_2 \cdot (R_d/R_g), \quad \text{PAR} = R_1 + R_2 \cdot (R_d/R_g), \tag{10}$$

then the maximum of GPP is reached at $(R_d/R_g)_{\text{max}} = -0.5(L_1/L_2 + R_1/R_2)$, estimated as the point where the parabola GPP = LUE $\cdot$ PAR has its maximum. The position of this maximum depends on the ratios $L_2/L_1$ and $R_2/R_1$. For a certain range of parameters, the maximum of parabola can be located at $R_d/R_g < 0.08$, which is below the minimum diffuse fraction measured at our sites and, therefore, not feasible for the latitudes we consider in this study. In this case, GPP monotonically

decreases when $R_d/R_g$ increases from $\sim 0.1$ to 1. Conversely, $(R_d/R_g)_{max}$ should be larger than $0.08 - 0.1$ for GPP to have a maximum under diffuse light. Note, that PAR dependences on $R_d/R_g$ at the middle latitudes are similar, with $R_1/R_2 \approx -1.5$ while for SMEAR I this ratio is $R_1/R_2 \approx -1.3$. From these estimates, $L_2 > L_1/1.2$ for the middle latitudes. Since $L_1$ is roughly the minimum value of LUE at $R_d/R_g \approx 0.1$ (clear sky), while $L_1 + L_2$ is the maximum of LUE at $R_d/R_g = 1$ (overcast conditions), $L_2$ can be treated as the maximal gain in LUE under diffuse light. Thus, GPP will have a maximum associated with diffuse radiation if the ecosystem LUE under diffuse light increases by more than approximately 80% of its minimum possible value (which is observed under clean conditions on clear days). For the sites considered in the present study, the smallest gain in LUE due to diffuse radiation is observed at SMEAR II, where LUE under diffuse light was almost twice as large as its value on clear days. The largest gain was at SMEAR Estonia and Fonovaya where LUE grew almost by a factor of 3 if the dominating radiation conditions in the area changed from mostly direct to mostly diffuse radiation. Therefore, all ecosystems displayed maxima of GPP dependence on $R_d/R_g$ due to diffuse light, though at different values of $R_d/R_g$.

Morever, this approach clearly demonstrates that the maximum of GPP can never be reached under overcast conditions. If we take again $R_1/R_2 =$ -1.4 as for the middle latitudes, then the position of the maximum is at $(R_d/R_g)_{max} = -1/2(L_1/L_2)$ +0.7. One can immediately deduce that for the large slopes of LUE, i.e. when $L_1/L_2$ approaches zero, $(R_d/R_g)_{max}$ approaches 0.7. At SMEAR I, this position is restricted by $(R_d/R_g)_{max} \approx 0.65$. The maximum of GPP dependence on $R_d/R_g$ for five sites considered in this study is at $(R_d/R_g)_{max} \approx 0.4 - 0.5$.

### 3.3 Discussion

In this section we combine the results from the previous sections to make the conclusions regarding direct effect of aerosol on GPP and compare obtained results with previous studies.

As was mentioned in section 3.1.3, a cloud-biased data set together with a standard linear regression analysis results in weak but significant ($p < 0.001$) correlations between CS and $R_d/R_g$ (Table 3). The relatively high cloud-biased diffuse fraction of global radiation at low CS leads to an underestimation of the effect of increasing aerosol loading for the cloud-biased data set. If CS increases from $0.002 \text{ s}^{-1}$ to $0.015 \text{ s}^{-1}$ (obtained for the clear sky conditions), a relative increase in the diffuse fraction of global radiation following from the clear-sky model is from 110% to 165% at all the sites except Fonovaya, while this increase is between 65% and 118% for cloud-biased data. In what follows we use only the results for clear-sky model (representing the measured data set when the stricter criterion (8) of clear sky is applied, as follows from Fig.3). In absolute values the increase was quite small: from 0.11 to $\sim 0.27$ at SMEAR I and II, from 0.11 to $\sim 0.2$ at SMEAR Estonia and Fonovaya. The increases in $R_d/R_g$ over these value ranges lead to increases in GPP from 17.2 to 18.6 $\mu$mol s$^{-1}$ m$^{-2}$ at Fonovaya, from 18.5 to 20.9 $\mu$mol s$^{-1}$ m$^{-2}$ at SMEAR Estonia, from 15.8 to 16.9 $\mu$mol s$^{-1}$ m$^{-2}$ at SMEAR II and from 8.8 to 10.0 $\mu$mol s$^{-1}$ m$^{-2}$ at SMEAR I. The largest relative increase in GPP due to the increasing aerosol loading from its minimum value to the maximum value was observed for SMEAR I and SMEAR Estonia (14% and 13% respectively). Note, however, that the median value of CS should be increased by a factor of about 5 at SMEAR I to get this maximum gain in GPP, whereas this same increase in GPP would be observed at SMEAR Estonia if the median CS increases by a factor of 2-3.

Overall, we obtained rather weak dependence of the diffuse fraction on CS. It is much weaker than that reported by Kulmala et al. (2014): for all the sites the slope is less than 10 s (Table 3) as compared to almost 100 s obtained in the above mentioned study. This difference is due to inappropriate criterion of clear sky selecting cloud-biased points with diffuse fraction up to 0.8 (Kulmala et al., 2014) and a different statistical method (bivariate fitting as compared to linear regression used in this study).

Note that Kulmala et al. (2014) reported minimum and maximum possible slopes for an increase in the diffuse fraction of global radiation with CS. Our present results are close to their minimum slope. Furthermore, due to the large diffuse fractions attributed to the effect of aerosol rather than clouds, the maximum direct effect of aerosol on GPP was overestimated by Kulmala et al. (2014). In the present study we obtained 6% increase in GPP due to the diffuse radiation effect rather than $\approx 30\%$ reported by Kulmala et al. (2014). However, their minimum slope, reported for GPP vs $R_d/R_g$ dependence, would

result in an increase of GPP similar to this study.

Note that aerosol loading observed at all sites corresponds to $0.04 < \text{AOD}_{675} < 0.35$ with the typical values being in the range 0.05-0.10 and $\text{AOD}_{500} < 0.25$ (see Appendix A). In accordance with the study by Park et al. (2018), an increase in diffuse fraction did not exceed 0.3 for these relatively low AOD values. Much higher diffuse fractions (0.5-0.7) due to the direct aerosol effect were obtained by Cirino et al. (2014) for biomass burning season at Amazon.

Next, all the GPP dependences have a maximum due to clouds. The maximum corresponds to the clouds with the diffuse fraction on the order of 0.4-0.5. According to Cheng et al. (2016) and Pedruzo-Bagazgoitia et al. (2017), this $R_d/R_g$ corresponds to optically thin clouds with cloud optical thicknesses less than 5. Conversely, GPP decreases for optically thick clouds, which has also been demonstrated by Cheng et al. (2016). The largest increase is 32-33% at SMEAR Estonia and Fonovaya, whereas the smallest increase is 11% at SMEAR II as compared to the GPP values on clear days characterized by

low aerosol loading. At the middle latitudes with the similar attenuation of radiation due to aerosols and clouds, the increase in GPP depends on the LUE slope: the steeper is LUE slope is, the more pronounced is the maximum.

It follows from Fig.8 that similar forest stands at Zotino and SMEAR I demonstrated similar dependencies of GPP on $R_d/R_g$, while this dependence was different for the coniferous forest at SMEAR II. GPP at SMEAR II under clear sky is almost 1.5 times larger than the corresponding GPP at SMEAR I and Zotino, but GPP increase under cloudy sky is smaller at

SMEAR II. This could be the consequence of the closed canopy and higher leaf area index of the SMEAR II forest stand. Our GPP data sets, reported in Fig.9, look similar to those reported by Alton et al. (2007) and Alton (2008). The GPP dependence reported for SMEAR II is also similar to that reported by Alton (2008) for needle-leaf forests, but for mixed forests we obtained increase up to 30% as compared to moderate 10% increase for broadleaf forests reported by Alton (2008). Note that he used parametrization, eq. (1), for the diffuse fraction of global radiation while we had measurements of diffuse radiation at four sites

out of five.

Finally, we considered the data from Zotino including the periods of forest fires (Park et al., 2018). Fig. 7 suggests that forest fires do not have any specific influence on PAR decrease with increasing $R_d/R_g$ as compared to cloudy sky. In other words, plumes from forest fires lead to similar decrease in PAR and similar separation in diffuse and direct fractions as some of clouds. The same holds for the LUE of an ecosystem: the dependence of LUE on $R_d/R_g$ at Zotino is similar to that of other coniferous

sites. However, a significant increase in GPP under wildfire plumes can be potentially obtained on the daily time scale because

the radiation regime with $(R_d/R_g)_{max}$, i.e. close to the optimal conditions for ecosystem photosynthesis, can persist for a long time under plume. At the same time, clouds may be intermittent and the effect of sporadic GPP increase can be compensated by the smaller GPP when clouds are in front of the sun and radiation is reduced (Pedruzo-Bagazgoitia et al., 2017).

## 4   Conclusion

We quantified the direct effect of aerosol on solar radiation and GPP in boreal and hemiboreal forests in Eurasia. The analysis was based on the data from five sites including coniferous and mixed forest ecosystems in Eurasia. The diffuse fraction of global radiation due to the direct aerosol effect was estimated to be in the range $0.11 < R_d/R_g < 0.27$ at all the sites.

For the first time we demonstrated a connection between solar radiation properties (the diffuse fraction of global radiation) and condensation sink. The latter parameter is used in aerosol studies and it is obtained from ground-based observations. Employing CS instead of a column-averaged aerosol parameter AOD is a necessary step towards further investigation of the COBACC climate feedback loop, linking biogenic volatile organic compounds emissions and aerosol characteristics.

The GPP-radiation analysis was performed using the separation of GPP into LUE and PAR. We found a linear dependency between the diffuse fraction of solar radiation and LUE, as well as between the diffuse fraction of solar radiation and PAR, for all the sites. While the PAR dependences were quite similar to each other (except for SMEAR I located at relatively high latitude), the LUE dependences were different: the slopes were 60% steeper for mixed forests than for coniferous forests and the intercepts were about 40% lower for coniferous forests with open canopies. We obtained a parabolic shape for the GPP dependence on the diffuse fraction of solar radiation. The maximum of the parabola was more pronounced for mixed forests due to the above-mentioned differences in the LUE dependences between the mixed and coniferous forests. Note that parabolic, or near parabolic, shapes have been reported for different forest sites also by Alton (2008) and Moffat et al. (2010) using other methods than were used in this study.

We showed that GPP can be increased by 6-14% due to the direct effect of aerosol particles at remote sites as compared to clean conditions with low values of CS. The maximum increase was observed for mixed forests at the mid latitudes and for coniferous forests at relatively high latitudes.

Furthermore, based on the similarity in the PAR dependences on the diffuse fraction of solar radiation for all the sites, we obtained the constraints on the ecosystems' LUE increase under diffuse light necessary for a GPP maximum due to diffuse light. At the mid latitudes, the LUE of an ecosystem should increase not less than by $\sim 80\%$ under diffuse light as compared to its value under clear sky conditions. Moreover, at the mid latitude sites, the diffuse fraction of solar radiation corresponding to the maximum GPP can not exceed 0.7.

The specific shape of the GPP dependence on the diffuse fraction of solar radiation suggests that clouds with half an hour-averaged fraction $R_d/R_g$ between 0.4 and 0.5 play an important role for ecosystems' GPP and demand further investigation. An increase in GPP due to clouds can reach 32-33% for mixed forests and 21-26% for coniferous forests with open canopy. Other relevant questions include cloud effects on the radiation regime and ecosystems' GPP on annual scale and investigation of potential aerosol effects on the evolution of clouds over forests.

*Data availability.* Data measured at the SMEAR II and SMEAR I stations are available on the following website: http://avaa.tdata.fi/web/smart/. The data are licensed under a Creative Commons 4.0 Attribution (CC BY) license. Other data sets can be available from the authors upon request.

## Appendix A: Condensation sink vs AOD

To make our study comparable to studies that use $AOD_{500}$ as a parameter quantifying aerosol loading, we performed an additional analysis for the data sets from SMEAR II and SMEAR Estonia. We used half-an-hour averaged AOD from Aeronet sites and CS, corresponding to local time 9.00-15.00 and maximal growing season. The data set from SMEAR II includes 3 years (2008-2010) and the data set from SMEAR Estonia includes 4 years (2013-2016).

The results are shown in Fig. A1. Though $AOD_{500}$ clearly increases with increasing CS, the scatter of data is great. It means
that in spite of the fact that both parameters are roughly proportional to aerosol surface area distribution (Sundström et al., 2015), and in spite of presumably well-mixed boundary layer during daytime, there is no simple relationship between ground-based and column-integrated aerosol characteristics. This has been also noted by Sundström et al. (2015) for sites at South Africa. Sundström et al. (2015) found a moderate correlation between in situ measured scattering coefficient and AOD from Aeronet. At the same time, scattering coefficient was strongly correlated with CS, therefore, a moderate correlation between
CS and AOD from Aeronet could be expected. Fig. A1 shows moderate correlation (R = 0.53) for boreal forests at the middle latitudes.

*Competing interests.* The authors declare that they have no conflict of interest.

*Disclaimer.* This study is a part of the Pan-Eurasian Experiment program (Lappalainen et al., 2016), and the results are reported from all the sites in Eurasia being partners of the PEEX program and performing continuous multidisciplinary measurements pertaining to boreal forests.

*Acknowledgements.* The authors are grateful to two referees that helped to improve clarity of the manuscript. This work was supported by the Academy of Finland Center of Excellence programme (grant no. 307331) and Academy of Finland professor grant to MK (no. 302958). The results are part of a project (ATM-GTP/ERC) that has received funding from the European Research Council (ERC) under the European Union's Horizon 2020 research and innovation programme (grant agreement no. 742206) and NordForsk via TRAKT-2018: Transferable Knowledge and Technologies for High-Resolution Environmental Impact Assessment and Management. SN and AK acknowledge
the support through the project the European network for observing our changing planet (ERA-PLANET, grant agreement 689443) under the European Union's Horizon 2020 research and innovation programme and the Estonian Ministry of Sciences project "Biosphere-atmosphere interaction and climate research applying the SMEAR Estonia research infrastructure" (grant P170026PKTF). Measurements at Fonovaya

station are performed under support of Russian Science Foundation (grant No 17-17-01095). Aerosol measurements at SMEAR Estonia were supported by the institutional research funding IUT20-11 and IUT20-52 of the Estonian Ministry of Education and Research.

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

**Table 1.** Data sets from different sites used in this study.

| Station | Parameters | Years |
|---|---|---|
| SMEAR I (67°46'N, 29°36'E, 390 m a.s.l.) | Global and diffuse radiation (PAR), particle number-size distribution, GPP | 2015-2016 |
| SMEAR II (61°51'N, 24°17'E, 181 m a.s.l.) | Global and diffuse radiation (broadband), particle number-size distribution, GPP | 2008-2009 |
| SMEAR II (61°51'N, 24°17'E, 181 m a.s.l.) | Global and diffuse radiation (PAR), particle number-size distribution, GPP | 2010,2014-2015 |
| SMEAR Estonia (58°16′N, 27°16'E, 36 m a.s.l.) | Global and diffuse radiation (broadband and hyperspectral), particle number-size distribution, GPP | 2015-2016 |
| Fonovaya (56°25'N, 84°04'E, 80 m a.s.l.) | Global radiation (short wave), particle number-size distribution, $CO_2$ concentration at 10 and 30 m, wind speed, pressure, air temperature, relative humidity | 2016-2017 |
| Zotino (60°48'N, 89°21'E, 180 m a.s.l.) | Global and diffuse radiation (PAR), $CO_2$ flux, air temperature | 2012-2016 |

**Table 2.** Best fit parameters for the diffuse fractions ratio, $\frac{fdif_{\mathrm{PAR}}}{fdif_{\mathrm{bb}}}$, as a function of diffuse fraction of broadband radiation, $fdif_{\mathrm{bb}}$, for various solar zenith angles (eq. (7)).

| Solar zenith angle | $a$ | $b$ | $c$ | $d$ |
|---|---|---|---|---|
| $35° < sza < 45°$ | 0.186 | 0.140 | 0.318 | 0.990 |
| $45° < sza < 55°$ | 0.191 | 0.146 | 0.296 | 0.990 |
| $55° < sza < 65°$ | 0.143 | 0.351 | 0.346 | 0.980 |

**Table 3.** Best fit parameters, correlation coefficients and $p$-values for radiation data ($R_{\mathrm{d}}/R_{\mathrm{g}} = k\mathrm{CS} + b$).

| Station | $k_{\mathrm{mod}}$, s | $b_{\mathrm{mod}}$ | $R_{\mathrm{mod}}$ | $p_{\mathrm{mod}}$ | $k_{\mathrm{meas}}$, s | $b_{\mathrm{meas}}$ | $R_{\mathrm{meas}}$ | $p_{\mathrm{meas}}$ |
|---|---|---|---|---|---|---|---|---|
| SMEAR I | 8.30 | 0.108 | 0.53 | $< 0.001$ | 6.73 | 0.176 | 0.18 | 0.0422 |
| SMEAR II | 10.21 | 0.092 | 0.69 | $< 0.001$ | 11.59 | 0.153 | 0.33 | $< 0.001$ |
| SMEAR Estonia | 6.39 | 0.094 | 0.60 | $< 0.001$ | 5.50 | 0.123 | 0.23 | $< 0.001$ |
| Fonovaya | 3.32 | 0.113 | 0.44 | $< 0.001$ | - | - | - | - |

**Table 4.** Linear regression coefficients for PAR and LUE at different sites: $\text{LUE} = L_1 + L_2 \cdot (R_\mathrm{d}/R_\mathrm{g})$, $\text{PAR} = R_1 + R_2 \cdot (R_\mathrm{d}/R_\mathrm{g})$.

| Station | $L_2, \dfrac{\text{mol}_{\text{CO2}}}{\text{mol}_{\text{photons}}}$ | $L_1, \dfrac{\text{mol}_{\text{CO2}}}{\text{mol}_{\text{photons}}}$ | $R_2, \mu\text{mol s}^{-1}\text{ m}^{-2}$ | $R_1, \mu\text{mol s}^{-1}\text{ m}^{-2}$ |
|---|---|---|---|---|
| SMEAR I | 0.0157 | 0.0062 | -944 | 1212 |
| SMEAR II | 0.0164 | 0.0098 | -1081 | 1480 |
| SMEAR Estonia | 0.0278 | 0.0094 | -1194 | 1608 |
| Fonovaya | 0.0238 | 0.0092 | -1085 | 1575 |
| Zotino | 0.0143 | 0.0058 | -1118 | 1548 |

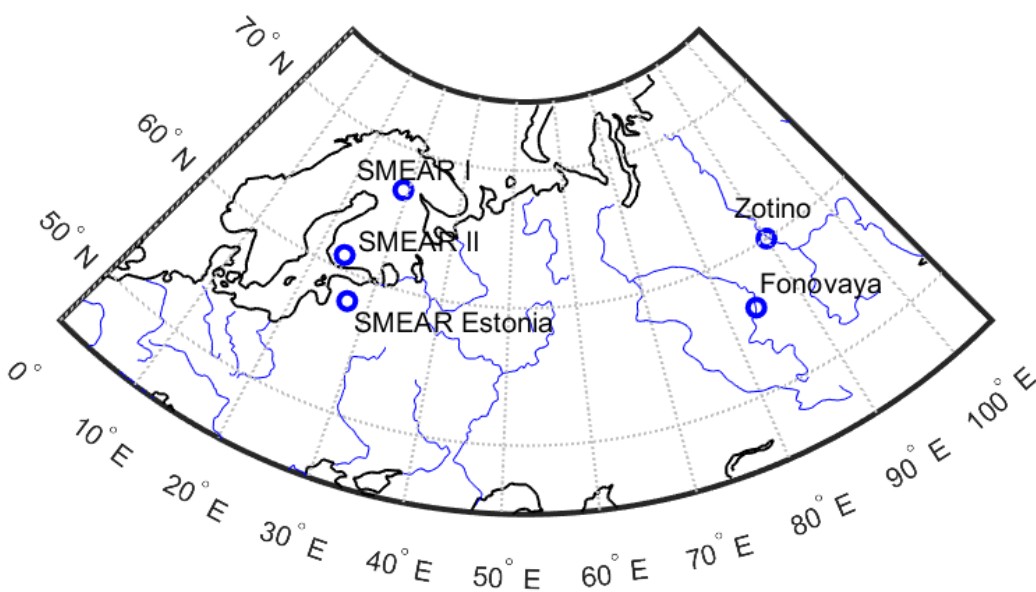

**Figure 1.** Location of the sites (see Table 1 for longitudes and latitudes of different stations).

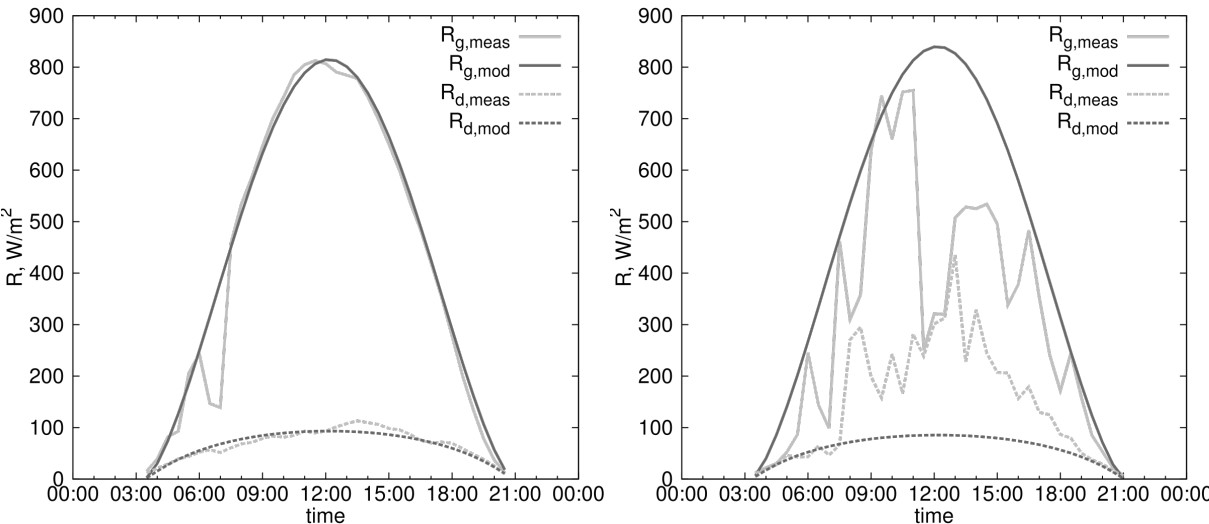

**Figure 2.** Modelled vs measured irradiance, SMEAR Estonia: measured global radiation $R_{g,meas}$, modelled global radiation $R_{g,mod}$, measured diffuse radiation $R_{d,meas}$, modelled diffuse radiation $R_{d,mod}$. Left panel: 01 June 2016, clear day; right panel: 06 June 2016, cloudy day. Time scale corresponds to local winter time (UTC+2).

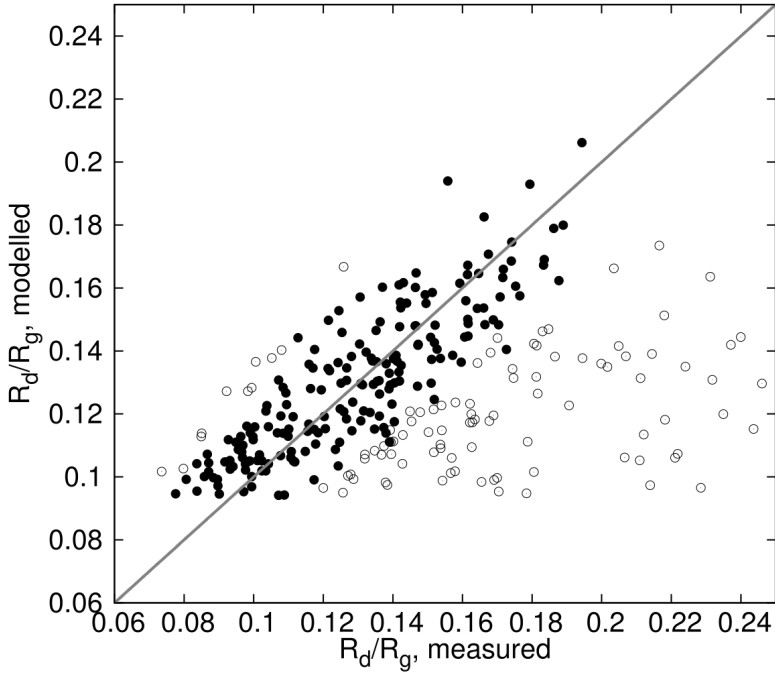

**Figure 3.** Diffuse fraction of global irradiance: modelled vs measured (SMEAR Estonia, 2016). Closed symbols: criterion of clear sky based on diffuse and global irradiance ($R_{g,meas}/R_{g,mod} > 0.9$, $R_{d,meas}/R_{d,mod} > 0.8$), closed symbols and open symbols: criterion of clear sky based on global irradiance ($R_{g,meas}/R_{g,mod} > 0.9$). The line illustrates the ideal ratio 1 : 1 between the modelled and measured data sets.

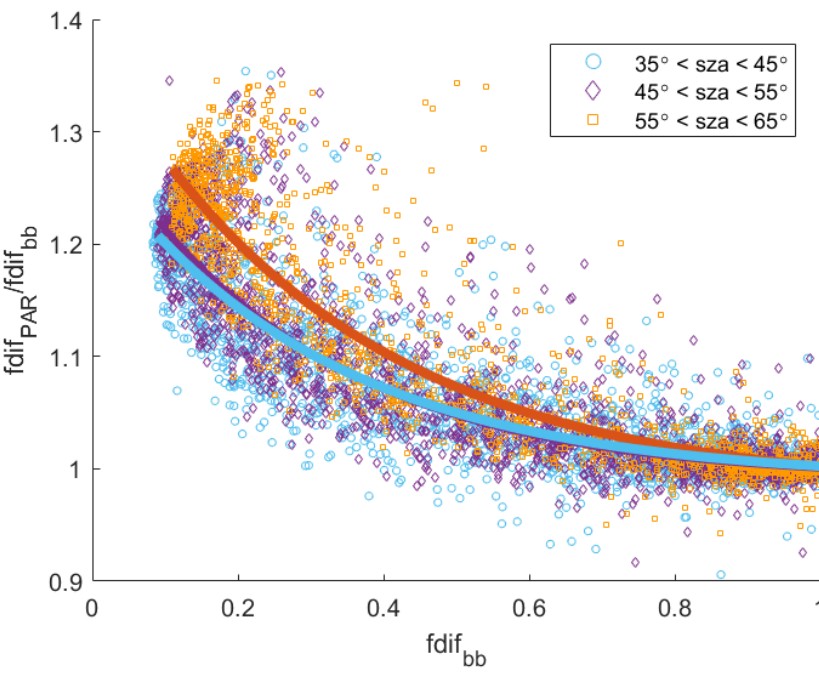

**Figure 4.** Ratio $fdif_{PAR}/fdif_{bb}$ as a function of $fdif_{bb}$ (SMEAR Estonia). Different curves correspond to the best fits of the data for different solar zenith angles $sza$.

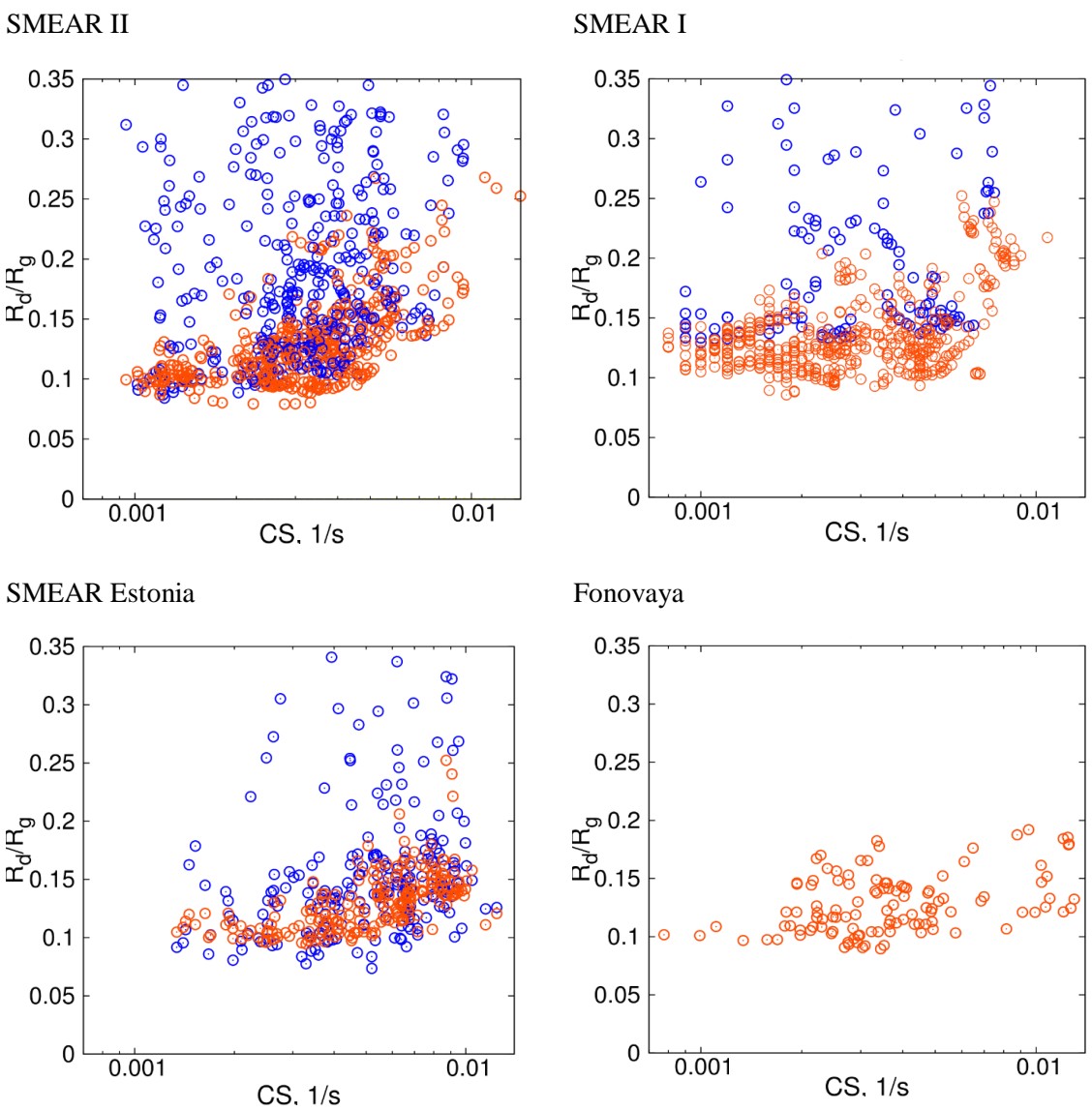

**Figure 5.** The diffuse fraction of global irradiance as a function of CS (clear sky data). Red symbols: calculations with the clear sky model, blue symbols: measurements.

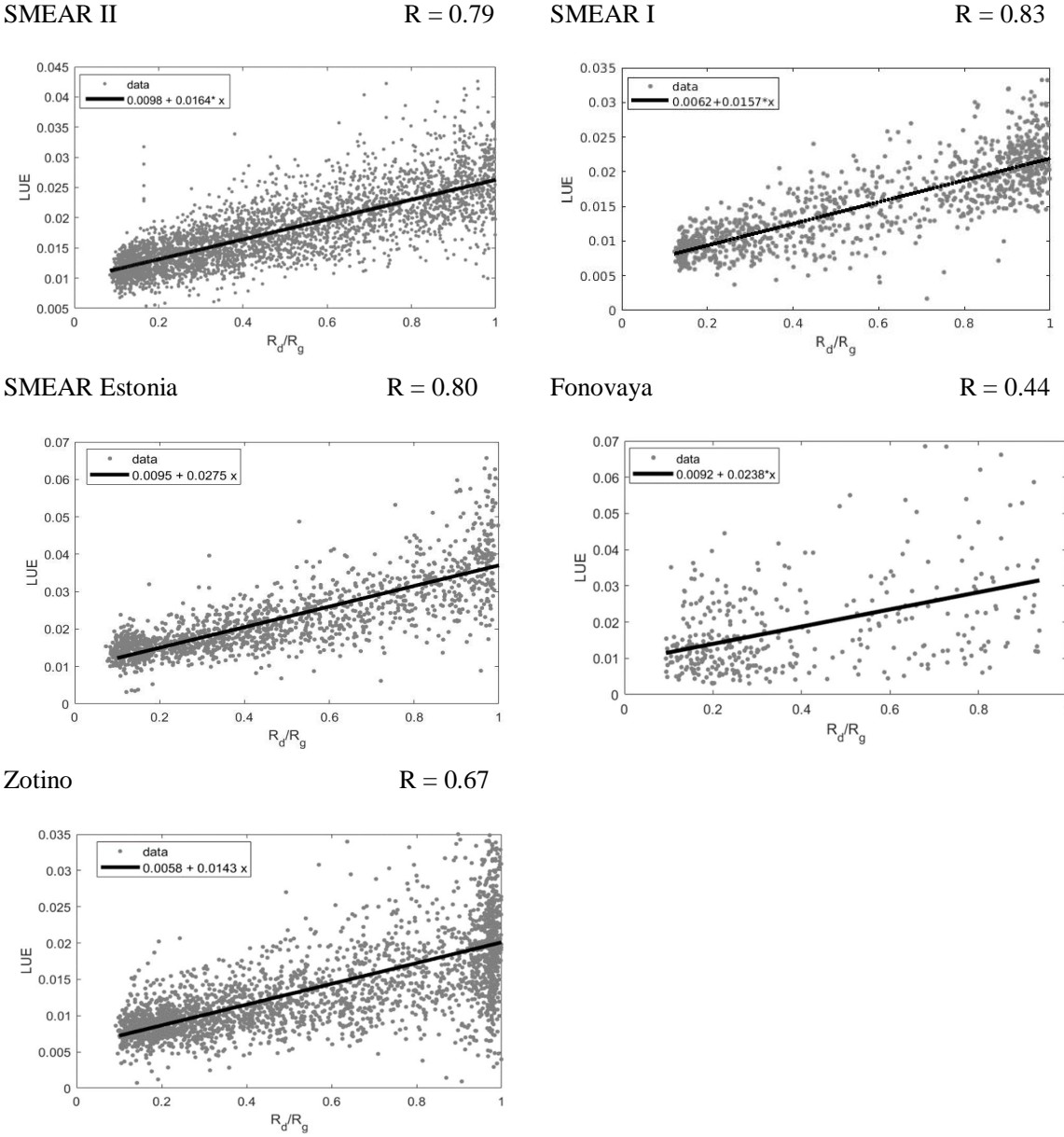

**Figure 6.** Light use efficiency (LUE) as a function of diffuse fraction of global irradiance ($R_d/R_g$). All dependences are statistically significant ($p < 0.001$).

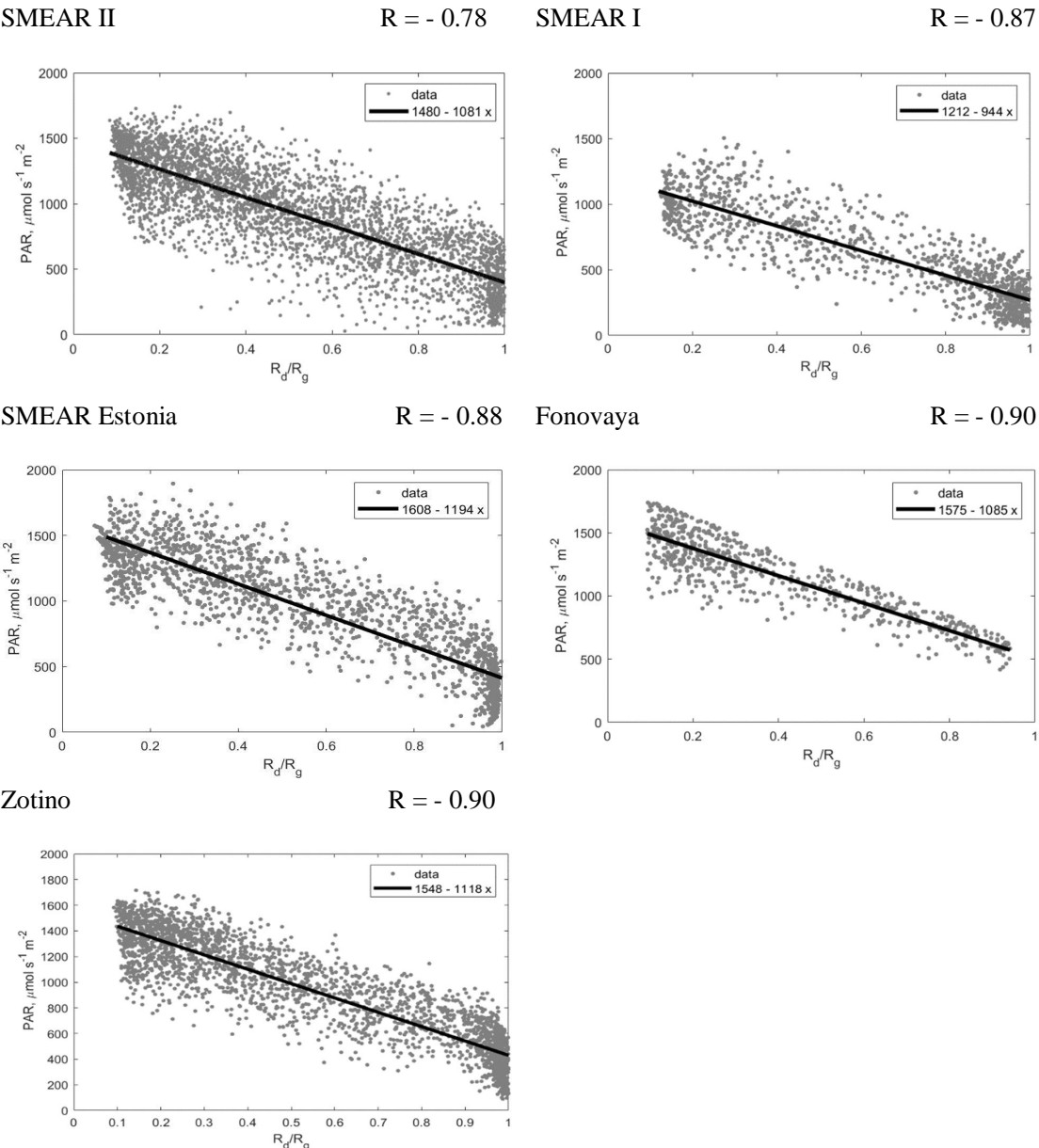

**Figure 7.** Photosynthetically active global radiation (PAR) as a function of diffuse fraction of global irradiance ($R_d/R_g$). All dependences are statistically significant ($p < 0.001$). The number of sample points is reported in the caption to Fig.6.

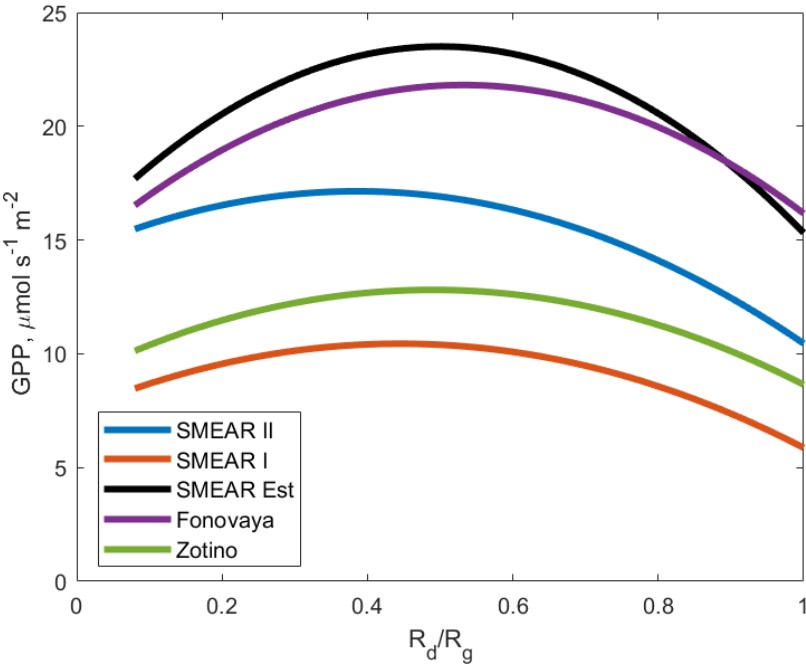

**Figure 8.** Estimated GPP dependences on $R_d/R_g$ for all the sites (obtained as GPP = LUE · PAR using the coefficients for PAR and LUE dependences on $R_d/R_g$ reported in Table 4).

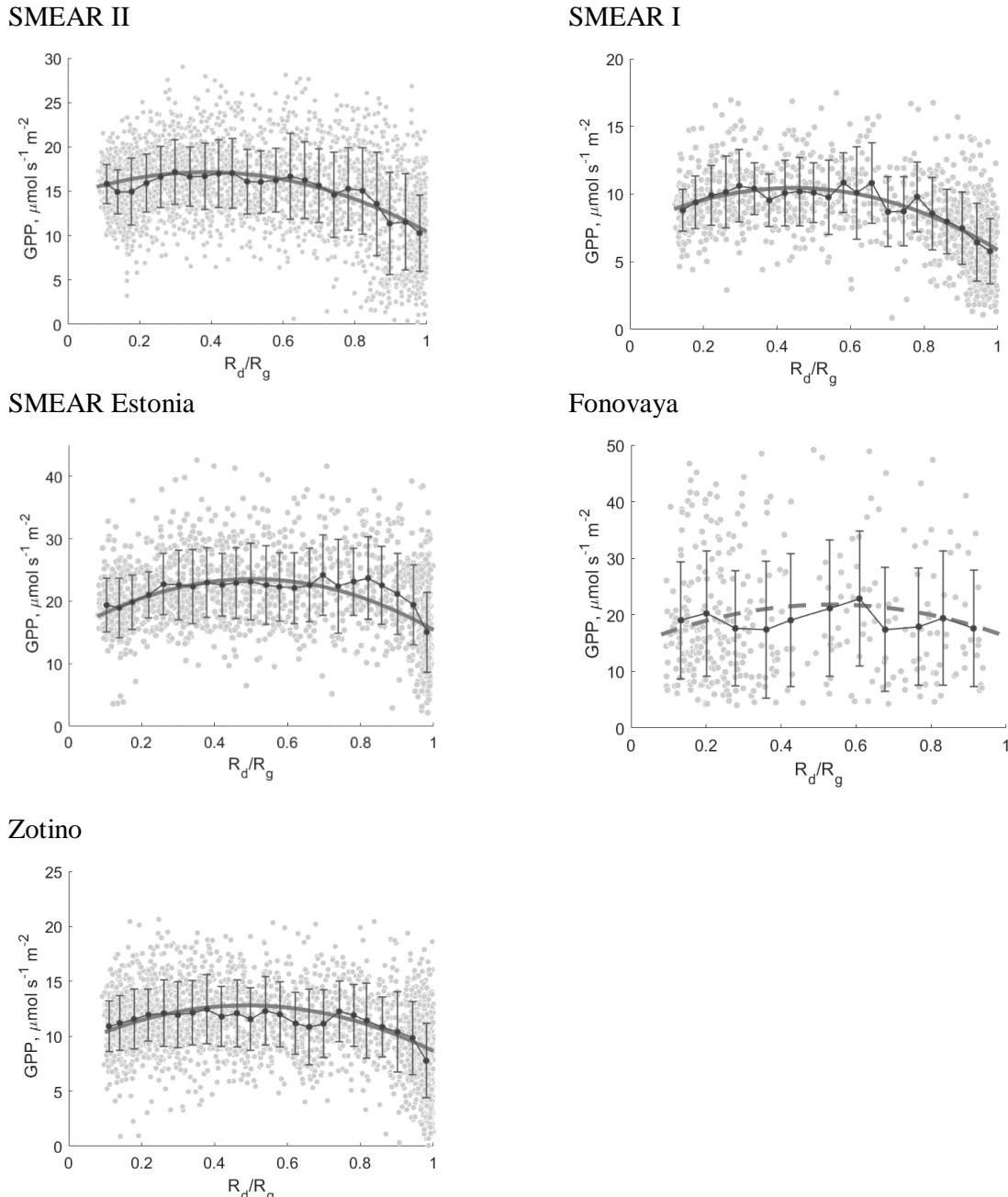

**Figure 9.** GPP dependences on $R_d/R_g$ for all the sites. The curves represent estimated GPP (the same parabolas as in Fig.8). We use dashed curve for Fonovaya because of the relatively low correlation coefficient obtained for LUE (R=0.44, Fig. 6). The data sets for all sites were cast in bins in $R_d/R_g$, the width of each bin is $R_d/R_g = 0.04$ ($R_d/R_g = 0.08$ for Fonovaya). Black points correspond to the mean GPP in each bin and error bars show the standard deviation of data for each bin.

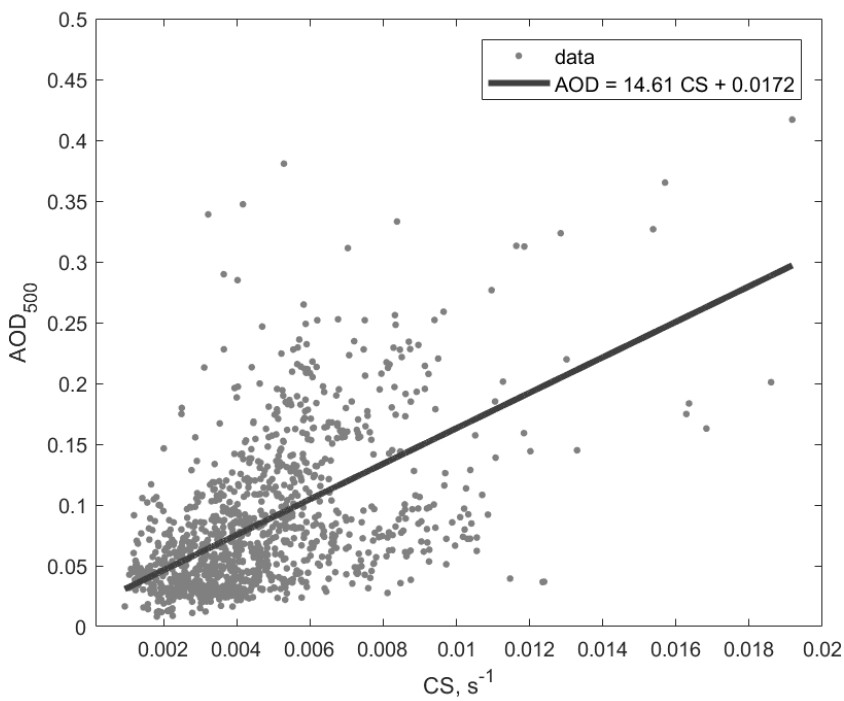

**Figure A1.** Aerosol Optical Depth at 500 nm (AOD$_{500}$) as a function of condensation sink (CS) for SMEAR II and SMEAR Estonia ($R = 0.53$, $p < 0.001$).