# Peer review of "Direct effect of aerosols on solar radiation and gross primary production in boreal and hemiboreal forests"

_Atmospheric Chemistry and Physics, 2018_

## Referee Comment (RC1) · Anonymous Referee #1 · 25 Aug 2018

The manuscript by Ezhova et al. estimates the impact of aerosols on solar radiation and GPP in five boreal ecosystems. This estimate is obtained from an analysis of data sets from these sites including information on GPP and direct and diffuse radiation as well as measured aerosol abundance (as particle number-size distributions). The authors find varying differences in the steepness of the response curves of GPP to diffuse fraction and optimum ecosystem responses at intermediate levels of diffuse radiation, meaning that enhanced aerosol load on clear days will give a positive response in GPP.

The subject of this study is highly relevant and interesting for ACP, and the data gathered and approach chosen by the authors provide a good basis for the study. However,

the analysis is sometimes hard to follow and difficult to judge, not least because the methods are described only very concisely and are largely intertwined with the results in sections 3 and 4. Moreover, the results are to some extent discussed in relation to sources of uncertainty, but there is little comparison of the authors' findings with other studies investigating aerosol effects on diffuse radiation or diffuse radiation effects on GPP. I would advise to improve and gather the description of the methods in a separate section, which would aid the reader in understanding the study, and to extend the comparison of the results in section 5 (or in a separate section) with other studies.

Apart from that, I have some concerns about the current analysis that would require more explanation from the authors. I provide my comments below, and would like to encourage the authors to improve the manuscript, as its results are very interesting for the research community.

Major remarks:

- The study lacks a Materials and Methods section. Section 2 provides basic information about the sites and table 1 provides a very concise summary of the variables that were used in the analysis, but the description of the data sets should be extended to be comprehensive. Please add basic information about the instrumentation used or references to papers that describe this, gaps/missing data in the data sets and possible gap-filling if applied. Also, many parts in sections 3 and 4 belong in a Materials and Methods section rather than in the Results: section 3.1 up to p. 5, l. 25, section 3.2 up to p. 7 l. 13 and p. 7 l. 22-25, section 3.3 up to p. 8, l. 24, section 4.1 up to p. 10, l. 23. Finally, sections 3 and 4 use more variables than table 1 does, sometimes with a concise description of their sources. I would suggest to add these (e.g., AOD700 and precipitable water (p. 5, l. 5)) also to the Materials and Methods section.

- Figure 2 and Solis modelling: The model result (Fig. 2b) does not seem to account for the cloudiness (magnitudes are roughly similar between Fig. 2a and Fig. 2b), is the model result here that of a clear day? Have the optical depths in Eq. 2 and 3 been

adjusted to the actual measured values? Or is this intentional and meant only to show that values at or above the clear-sky expected value can be found even at cloudy days?

- p. 7, l. 16: How is this number determined? Which measurements were used for this, which time of year, etc?

- p. 9, l. 3: I do not agree with the "observed ... increase in Rd/Rg with increasing CS": The model seems indeed to show this, but the three sites with observations show a huge spread and no clear correlations. Please test statistically whether there is a relationship between Rd/Rg and CS - I have clear doubts about that. See also the low correlation coefficients (l. 12-14) - R2 values are extremely low, so I guess that these results are not significant.

- p. 10, l. 13: Where does the 0.8 in the absorption come from? This number depends on (amongst others) LAI. Later, the LAI is used as an argument for differences in LUE, whereas I would rather assume that it affects PARabs.

- p. 11, l. 3: Where does the slope of 1130 W m-2 and its uncertainty (5%) come from? In figure 7, PAR is given in umol m-2 s-1, has this been converted?

- Fig. 8 and p. 11, l. 15: The optimum curves are very interesting, but where are the data in these curves? It would be interesting to see how well these curves can capture the actual observations, rather than only using the two linear relationships obtained from the observations before. It would also give an impression of how uncertainties propagate, and it may even be interesting to apply the same separation between clear and cloudy days as done in Fig. 3 and 5 to show how well these relationships work for each of the two types.

- p. 11, l. 17: For interpreting the GPP curves with the aerosol data, it should be noted that the aerosol analysis in section 3 has focused on clear days with conditions of Rd/Rg < 0.25, whereas the GPP analysis focuses on the entire range (including clouds). Please acknowledge this in the discussion of the results: The discussed variations in CS (l. 18) are all for clear days only.

- p. 12, l. 18: Why is such a low positioning of the optimum not feasible for these latitudes? This would simply mean that the decreasing PAR has a stronger impact than the increasing diffuse fraction with more aerosols, right?

Minor remarks:

p. 4, l. 18: Please explain what Aeronet sites are, or generalize the statement about availability of data from nearby sites.

p. 7, l. 4: Please provide a reference for the wavelengths that are affected.

p. 7, l. 5-10: This paragraph is hard to follow. If I understand it correctly, the authors want to state that aerosols interact more pronouncedly with PAR wavelengths (400-700 nm - the range could be mentioned to clarify the sentence) than with NIR wavelengths, so that the amount of diffuse PAR is relatively larger than diffuse global radiation or diffuse NIR. Correct?

p. 7, l. 18: I miss the logic in this sentence: Why are wavelength-sensitive interactions more pronounced with lower amounts of diffuse radiation?

p. 7, l. 29: You could replace x and f(x) in the equation with the respective parameters (fdifbb and fdifPAR/fdif)

p. 7, l. 31: Please add unit of the PAR quantum efficiency.

p. 8, l. 25: replace "is" by "are"

p. 9, l. 9: Please provide a reference for the low scattering for CS<0.005 s-1.

p. 10, l. 23: Where does the "increase or decrease" come from? Generally, lower light levels would give a relatively better usage of the light because saturation is not reached (meaning a higher LUE). Are there conditions where you would expect a decrease instead?

p. 12, l. 1: Is this analysis of forest fire impact shown anywhere? Fig. 7 does not separate between forest fire and non-forest fire days.

p. 13, l. 9: Why can AOD not be used for estimating the feedback loop?

---

## Referee Comment (RC2) · J. Vila (Referee) · 4 Sep 2018

The authors present and discuss a very interesting study on how the diffuse radiation driven by the presence of aerosol enhances gross primary production (GPP). Based on observations taken at five sites, they reported an increase that varies between 6-14% depending on the aerosol loading. In the methods, special attention is paid to separate the effect of clouds from aerosol, and to determine the differences between homogeneous coniferous forest at high latitudes and mixed forest at lower latitudes. This is done by applying a criteria in the observations to determine the effect of diffuse radiation on the light use efficiency and the photosyntetically active radiation. I find
the results very interesting and worth to be published in Atmospheric Chemistry and Physics. I include my comments on potential improvements of the paper.

Comments:

1- In the complete and very-well written introduction, they use a very general terms from clouds. I believe it will be interesting to mention to the reader than thin clouds (with cloud optical depths below ~5) have a different impact on GPP than thick clouds (lines 1-5 in page 2) (see Pedruzo-Bagazgoitia et al, 2015).

2- I understand that the authors opted for a simple radiative transfer model due to a more complex radiative transfer model will require more input information that maybe is not available. My question here is if they have a reference on a study on how this simplification of the transfer of radiation might influence their findings.

3- I also understand that they employ irradiances in their analysis (Eqs. 2-4 at page 5). Here, I would like to hear the opinion (or a discussion point) of the authors if the actinic flux can be a better variable to determine the effect of aerosol on GPP.

4- Perhaps, and in order to make connections with other studies, it is worth to show every now and then an equivalence between the condensation sink and the aerosol optical depth. Closely related to this, how relevant is the scattering efficiency (line 15 page 8) as an independent variable from the condensation sink in their study?

5- A general comment that it might be relevant. I miss in all the Figures information on the canopy height. In my opinion, this information should be given due to the different transmissivities of direct and diffuse radiation in the canopy. For instance, in figures 6 and 7, they could give different colours at which heights the measurements were taken. To be more comparable, this could have been done normalized by the canopy height.

6- Could the authors explain better the overestimation of the cloud-biased data? (line 15 page 9)

7- I believe their criteria is robust to distinguish between aerosol effects and thin clouds

(line 35 and p[age 10). However, haze can be very difficult to distinguish. Could the authors comment on this point?

8- Figure 8 summarizes and it is in my opinion the highlight of the paper. However, all the data is gone and only the estimated dependences are given. Why? I understand that the data can be very scattered but I think it can be interesting for the reader to see by him/herself these maximum behaviour. The behaviour reminds me the one reported by Min and Wang (Geophysical Research Letters oi:10.1029/2007GL032398, 2008, see Figure 1). Since they don't have a discussion section, I think as a reader I will appreciate a more elaborate discussion.

---

## Author Comment (AC1) · 12 Nov 2018

**Replies to the comments of Referee #1**

We are grateful to the referee for the constructive criticism, which helped to improve the clarity of the manuscript. Please find below the replies to the specific comments and an account of the modifications implemented.

1. *The study lacks a Materials and Methods section. Section 2 provides basic information about the sites and table 1 provides a very concise summary of the variables that were used in the analysis, but the description of the data sets should be extended to be comprehensive. Please add basic information about the instrumentation used or references to papers that describe this, gaps/missing data in the data sets and possible gap-filling if applied. Also, many parts in sections 3 and 4 belong in a Materials and Methods section rather than in the Results: section 3.1 up to p. 5, l. 25, section 3.2 up to p. 7 l. 13 and p. 7 l. 22-25, section 3.3 up to p. 8, l. 24, section 4.1 up to p. 10, l. 23. Finally, sections 3 and 4 use more variables than table 1 does, sometimes with a concise description of their sources. I would suggest to add these (e.g., AOD700 and precipitable water (p. 5, l. 5)) also to the Materials and Methods section.*

   The original manuscript has been organized in two big parts following the main topics of the study: aerosol-radiation interaction and radiation-photosynthesis interaction.

   Following the reviewer's comments, we resort to the more common structure in this paper. We re-organised the manuscript, introducing Methods and Discussion as separate sections. Note that the text has not been changed significantly, rather we added links and small discussions to make it more understandable for the reader in a new form. Wherever we added the new text, we colored it in blue to make it easier for the reviewer to follow.

   The references to papers that describe the instrumentation were already in the text (p. 3, l. 30-31). We added the discussion on gap-filling in p. 4, l. 33-34 and p. 5, l. 1-2, 11-13. We compare the results with the studies of other authors in p. 14, l. 1-14 and l. 25-30.

2. *Figure 2 and Solis modelling: The model result (Fig. 2b) does not seem to account for the cloudiness (magnitudes are roughly similar between Fig. 2a and Fig. 2b), is the model result here that of a clear day? Have the optical depths in Eq. 2 and 3 been adjusted to the actual measured values? Or is this intentional and meant only to show that values at or above the clear-sky expected value can be found even at cloudy days?*

   As a clear sky model, Solis can not be used to model cloudy sky conditions. However, if there are patchy clouds, i.e. there are time moments with clear sky, Aeronet data are available for this day. They can be used to model what would be on this day, with this particular AOD and PW, if the day was clear. The reviewer is correct: we deliberately show this figure to illustrate that global radiation values close to the clear sky values can be reached on cloudy days and therefore, criteria based on the proximity of measured and modelled global radiation would choose these points as clear-sky points. However, diffuse radiation values are significantly higher for these points, which is due to clouds - averaged over half-an-hour data will be almost inevitably affected by clouds in case of patchy clouds. We added clarifying sentences in p. 9, l. 9-13.

3. *p. 7, l. 16: How is this number determined? Which measurements were used for this, which time of year, etc?*

The PAR quantum efficiency for SMEAR II was determined using measurements of PAR with Li-Cor Li-190SZ and measurements of broadband radiation with pyranometer Middleton SK08. The data were averaged over half-an-hour, and correspond to 9.00-15.00 local time, growing season. We divided global PAR by global broadband radiation and found a mean value, which is reported as quantum efficiency (we added this information in the manuscript: p. 6, l. 27-30). Measurements of PAR and global radiation reported by Ross and Sulev (Sources of errors in measurements of PAR, Agricultural and Forest Meteorology, 100, 103 - 125, 2000) were also performed during growing season with Li-Cor Li-190SA and a typical pyranometer (the model is not reported, but the measurements give results close to 'ideal' in the range of wavelengths from 300 nm to 2500 nm). Ross and Sulev (2000) apparently used the data from the measurements during the growing seasons 1994-1996 in Estonia. At least part of the explanation for the difference in the numbers is probably in the sensor for the global radiation measurements at SMEAR II. A data from the new sensor, used in 2016 near SMEAR II, result in the PAR quantum efficiency 1.89, closer to the value reported by Ross. However, since we work with the SMEAR II data, we used our value.

4. *p. 9, l. 3: I do not agree with the "observed ... increase in Rd/Rg with increasing CS": The model seems indeed to show this, but the three sites with observations show a huge spread and no clear correlations. Please test statistically whether there is a relationship between Rd/Rg and CS - I have clear doubts about that. See also the low correlation coefcients (l. 12-14) - R2 values are extremely low, so I guess that these results are not signicant.*

We apologise if this has not been written clear enough. In Fig. 5 we show measured and modelled data. The data are chosen using a simple criterion (7) of clear sky, based on the comparison between measured and modelled global radiation. Our aim was to demonstrate that this simple criterion is not good enough for the purposes of our study (quantification of the diffuse fraction due to aerosol) and also to demonstrate the consequences of having an inappropriate criterion.

First, the badness of criterion (7) has been demonstrated in Fig. 2. We show that for the points, selected by this criterion on cloudy days, diffuse radiation is increased as compared to what would be obtained from clear sky model using aerosol and precipitable water parameters measured on this day.

Second, Solis has shown good results for global and diffuse radiation under clear sky both in the US (Sengupta and Gotseff, Sengupta, M. and Gotseff, P.: Evaluation of Clear Sky Models for Satellite-Based Irradiance Estimates, Tech. Rep. NREL/TP-5D00-60735, National Renewable Energy Laboratory, 2013) and at our sites. Therefore, the results of the model can be used for quantification of the diffuse fraction. Furthermore, Fig. 3 demonstrates that for more advanced clear sky criterion (8), using comparison between measured and modelled diffuse and global radiation, also measured and modelled diffuse fractions are well correlated (closed circles). At the same time, open circles in Fig. 3 represent a big fraction of points, additionally selected by the simple criterion (7), which have high diffuse radiation, meaning that they are biased by clouds, similar to the points at Fig. 2b.

Therefore, in Fig. 5 model predictions for the diffuse fraction should be treated as would-be clear sky data. According to Fig. 3, we would get the same results also for measured data if the appropriate criterion of clear sky is used, since in this case measured and modelled diffuse fractions are well correlated. At the same time, measured points demonstrate the consequence of having an inappropriate criterion of clear sky. As the reviewer has noted, the correlations obtained for measured data are very weak, while modelled data clearly shows moderate correlation.

5. *p. 10, l. 13: Where does the 0.8 in the absorption come from? This number depends on (amongst others) LAI. Later, the LAI is used as an argument for differences in LUE, whereas I would rather assume that it affects PARabs.*

The reviewer is correct that the absorbed fraction of PAR, fAPAR, depends on LAI. fAPAR in this study is included in LUE. We defined LUE, following Cheng et al. (Using satellite-derived optical thickness to assess the inuence of clouds on terrestrial carbon uptake, JGR: Biogeosciences, 2016), as LUE = GPP/PAR. It might indeed be important for analysis to separate LUE and fAPAR. However, to use LAI in our analysis, we would need to define it in a consistent way and, importantly, use the same measurement technique at all the sites. This is unfortunately not feasible. To get the estimate of fAPAR, we used the dependence of fAPAR on LAI from T. Majasalmi doctoral thesis as well as from the study by Hovi et al. (2016). The first shows that fAPAR levels off to 0.8-0.9 for LAI > 2.5, while the second provides similar numbers for tree height larger than 10 m. We added this discussion in the manuscript (p. 8, l. 10-14).

6. *p. 11, l. 3: Where does the slope of 1130 W m-2 and its uncertainty (5%) come from? In figure 7, PAR is given in umol m-2 s-1, has this been converted?*

Th slope should be in umol m-2 s-1. We corrected this in the text. The number 1130 has been initially estimated as an average value between maximum and minimum values at four sites (all except SMEAR I) and the deviation 5% - as the difference between the average and the minimum and maximum values of the slope reported in Table 4. However, to avoid misunderstanding, it now reads (p. 12, l. 7-8) "the slopes of the linear dependences in Fig. 7 were similar to each other (within the range 1081-1194 umol m-2 s-1)...".

7. *- Fig. 8 and p. 11, l. 15: The optimum curves are very interesting, but where are the data in these curves? It would be interesting to see how well these curves can capture the actual observations, rather than only using the two linear relationships obtained from the observations before. It would also give an impression of how uncertainties propagate, and it may even be interesting to apply the same separation between clear and cloudy days as done in Fig. 3 and 5 to show how well these relationships work for each of the two types.*

The optimum curves in Fig. 8 were supposed to give an impression of how the data on the sites are comparable to each other. Adding the data into this figure would hide the curves. We agree that it would be interesting for the reader to see the data and add one figure with five GPP panels (Fig. 9), similarly to LUE and PAR (Figs 6 and 7). Separation in clear and cloudy data will split the parabolas in two parts and clear sky part will be short (an example of such dependence can be found in Kulmala et al., CO2-induced terrestrial climate feedback mechanism: From carbon sink to aerosol source and back, Boreal Env. Res., 19B, 122-131, 2014). Generally, the increase will be seen there, but it will be more difficult to quantify.

8. *p. 11, l. 17: For interpreting the GPP curves with the aerosol data, it should be noted that the aerosol analysis in section 3 has focused on clear days with conditions of Rd/Rg < 0.25, whereas the GPP analysis focuses on the entire range (including clouds). Please acknowledge this in the discussion of the results: The discussed variations in CS (l. 18) are all for clear days only.*

We changed the sentence in the manuscript (p. 13, l.23): "If CS increases from 0.002 s$^{-1}$ to 0.015 s$^{-1}$ (obtained for the clear sky conditions)...".

9. *p. 12, l. 18: Why is such a low positioning of the optimum not feasible for these latitudes? This would simply mean that the decreasing PAR has a stronger impact than the increasing diffuse fraction with more aerosols, right?*

   This low positioning falls below the minimum ratio of diffuse to global radiation which can be measured. It is defined by Rayleigh scattering on air molecules which is always there and is responsible for the diffuse fraction at least 8% as we could see from the data corresponding to the cleanest atmosphere conditions. For different latitudes this minimum value can be slightly different due to the changing air mass. We added the clarifying sentence in the manuscript (p. 12, l.32-33).

10. *p. 4, l. 18: Please explain what Aeronet sites are, or generalize the statement about availability of data from nearby sites.*

    We changed this sentence in the manuscript, now it reads (p. 5, l. 15-19): We used AOD at 675 nm and PW from Aeronet (Holben et al.,1998); in particular, from the following Aeronet sites: Hyytiala (for SMEAR II), Sodankyla (for SMEAR I), Tomsk22 (for Fonovaya) and Toravere (for SMEAR Estonia). Aeronet sites are in immediate vicinity of Fonovaya and SMEAR II stations, SMEAR Estonia is 50 km away from Toravere and SMEAR I is approximately 70 km away from Sodankyla.

11. *p. 7, l. 4: Please provide a reference for the wavelengths that are affected.*

    The estimates for wavelengths are obtained from the previous sentence stating that "The characteristics of aerosol distribution become important for solar irradiance if $\pi d_p/\lambda \sim 1$", so one has to describe aerosol effect on solar radiation using Mie theory instead of Rayleigh scattering (e.g., Seinfeld and Pandis, 2016).

12. *p. 7, l. 5-10: This paragraph is hard to follow. If I understand it correctly, the authors want to state that aerosols interact more pronouncedly with PAR wavelengths (400-700 nm - the range could be mentioned to clarify the sentence) than with NIR wavelengths, so that the amount of diffuse PAR is relatively larger than diffuse global radiation or diffuse NIR. Correct?*

    Yes. The amount of diffuse PAR is expected to be relatively larger than diffuse broadband radiation or diffuse NIR. We added the range 400-700 nm in the sentence (p. 7, l. 9).

13. *p. 7, l. 18: I miss the logic in this sentence: Why are wavelength-sensitive interactions more pronounced with lower amounts of diffuse radiation?*

    We apologise for not being clear. We have reorganised and rewritten two paragraphs before this sentence (p. 6. from l. 24 and p. 7, till l. 12) and deleted this sentence. It was meant that wavelength-sensitive interactions will be more noticeable for diffuse radiation, the latter is relatively low as compared to global radiation (under clear sky and in clean atmosphere).

14. *p. 7, l. 29: You could replace x and f(x) in the equation with the respective parameters (fdifbb and fdifPAR/fdif)*

    We changed it in the manuscript (p.10, eq. (9)).

15. *p. 7, l. 31: Please add unit of the PAR quantum efciency.*

    We removed this sentence as this quantity was irrelevant for the study.

16. *p. 8, l. 25: replace "is" by "are"*

    We changed it in the manuscript (p. 10, l. 30).

17. *p. 9, l. 9: Please provide a reference for the low scattering for CS<0.005 s-1.*

    This follows from Fig. 5 and we gave the reference to this figure in the manuscript (p. 11, l. 14).

18. *p. 10, l. 23: Where does the "increase or decrease" come from? Generally, lower light levels would give a relatively better usage of the light because saturation is not reached (meaning a higher LUE). Are there conditions where you would expect a decrease instead?*

    Under diffuse light conditions (diffuse fraction close to one), the scatter in measured PAR data is also significant, with the highest values exceeding the saturation threshold for photosynthetic rate and the lowest values approaching zero. Therefore, LUE can change in a very wide range, and this is what we meant. Another plausible reason for an increased scatter in LUE is a relatively poor sensitivity of radiation sensors at low radiation levels. We changed this sentence in the manuscript to read (p. 8, l. 28-29): "Below this critical $R_{\mathrm{g}}$, LUE shows significant scatter (being high for the low radiation values); therefore we excluded these data from analysis."

19. *p. 12, l. 1: Is this analysis of forest re impact shown anywhere? Fig. 7 does not separate between forest re and non-forestfire days.*

    We just meant here that PAR dependence on the diffuse fraction is not affected anyhow by fire-influenced data: similar slope, similar scatter, similar uncertainties as compared to other places. In other words, we do not see a different separation of global radiation into direct and diffuse parts as compared to some types of clouds. This can be also concluded from Park et al.(Strong radiative effect induced by clouds and smoke on forest net ecosystem productivity in central Siberia, Agricultural and Forest Meteorology, 250-251, 376 - 387, 2018) analysis: data from her Fig. 7 fit well in the data set presented in our Fig. 7 for Zotino.

20. *p. 13, l. 9: Why can AOD not be used for estimating the feedback loop?*

    For global modelling and satellite analysis AOD would be more suitable. But for establishing fundamental links between ecosystem-related parameters (like BVOC) and aerosol, near-surface characteristics based on the measured aerosol properties would do a better job. In this study, we made an attempt to quantify this link between near-surface (CS) and column (AOD) aerosol properties. The parameters show moderate correlation (R=0.53), as one can see from Fig. A1 in the manuscript. At least for the present quantification of the loop, using ground-based measurements, we resort to CS, which is a local characteristic of aerosol near the ground, and could be later connected to measured BVOC properties (which will be the subject of a separate study).

We thank again the referee for the useful suggestions. We hope that the manuscript is now suitable for publication in ACP.

---

## Author Comment (AC2) · 12 Nov 2018

**Replies to the comments of Prof. J. Vila**

We are grateful to the referee for the constructive criticism, which helped to improve the clarity of the manuscript. Please find below the replies to the specific comments and an account of the modifications implemented.

1. *In the complete and very-well written introduction, they use a very general terms from clouds. I believe it will be interesting to mention to the reader than thin clouds (with cloud optical depths below 5) have a different impact on GPP than thick clouds (lines 1-5 in page 2) (see Pedruzo-Bagazgoitia et al, 2015).*

   We added the discussion on cloud thickness, but in our opinion, it belongs more to Discussion. It now reads (p.14, l. 15-18): "The maximum corresponds to the clouds with the diffuse fraction on the order of 0.4-0.5. According to Cheng et al. (2016) and Pedruzo-Bagazgoitia et al. (2017), this $R_d/R_g$ corresponds to optically thin clouds with cloud optical thickness less than 5. Conversely, GPP decreases for optically thick clouds, which has also been demonstrated by Cheng et al. (2016)."

2. *I understand that the authors opted for a simple radiative transfer model due to a more complex radiative transfer model will require more input information that maybe is not available. My question here is if they have a reference on a study on how this simplication of the transfer of radiation might inuence their ndings.*

   For the aims of this study, it is enough to have a reliable clear sky model, because the criterion of clear sky is based on the comparison between measured data and modelled clear sky radiation. The study showing how the simplified model performs in comparison with the full radiative transfer model is one by Ineichen (2008), and the comprehensive study demonstrating the validity of this clear sky model based on the measurements from several US sites was done by Sengupta and Gotseff (2013). More detailed discussion was given in the manuscript (p. 9, l. 12-13 and l. 25-31).

3. *I also understand that they employ irradiances in their analysis (Eqs. 2-4 at page 5). Here, I would like to hear the opinion (or a discussion point) of the authors if the actinic flux can be a better variable to determine the effect of aerosol on GPP.*

   Actinic flux represents spherically integrated energy on a volume of air, while irradiance represents the energy transported across a surface (Madronich, Photodissociation in the Atmosphere..., JGR, 1987). Therefore, irradiance is dependent on the incidence angle and decreases under glancing angles, while actinic flux remains constant. Thus, for a clear day, actinic flux would not change much in the range of moderate zenith angles. It would be changed presumably by clouds and aerosol presence, in contrast to irradiance, which additionally depends on the cosine of the solar zenith angle.

   The potential advantage of using actinic flux for daytime and maximum growing season GPP studies, similar to ours, is that GPP saturates after a certain radiation threshold (ca. 700-800 umol s-1 m-2) and stays relatively constant, similar to actinic flux (this does not account for wapor pressure deficit cycle, leading to higher GPP in the first half of the day). Thus, elimination of angle dependence could ideally keep both GPP and radiation parameter constant under clear sky conditions. One important reason to use irradiance is that this is the typically measured and

reported parameter which allows for comparison with other studies without additional confusion. This is relevant accounting for the fact that actinic flux is typically associated with atmospheric chemistry and UV-range of wavelengths.

4. *Perhaps, and in order to make connections with other studies, it is worth to show every now and then an equivalence between the condensation sink and the aerosol optical depth. Closely related to this, how relevant is the scattering efficiency (line 15 page 8) as an independent variable from the condensation sink in their study?*

   We agree with the comment, and add a figure demonstrating the connection between AOD500 and CS in the manuscript (see Appendix A and Fig. A1). Scattering coefficient and AOD characterize in situ and column-integrated scattering properties of aerosol respectively, we added a short discussion on this in Appendix.

5. *A general comment that it might be relevant. I miss in all the Figures information on the canopy height. In my opinion, this information should be given due to the different transmissivities of direct and diffuse radiation in the canopy. For instance, in gures 6 and 7, they could give different colours at which heights the measurements were taken. To be more comparable, this could have been done normalized by the canopy height.*

   Transmissivities of radiation depend not only on canopy height but also on LAI and on the distribution of leaves in the canopy, there is also difference for opened and closed canopies (e.g., Ross, 1981, The radiation regime and architecture of plant stands). We gave the information about the canopy height at the sites in Section 2. The measurements of LAI are not available for all sites. Moreover, this parameter is sensitive to the method of measurements and varies greatly even for the same site, which makes its usage difficult. For example, all-sided LAI of all trees >1 cm diameter in 2014-2015 from allometric equations (regression foliage biomass on tree dimensions) and measured average foliage area to mass ratios: SMEAR II - 7.3 $m^2$ $m^{-2}$, SMEAR I - 3.2 $m^2$ $m^{-2}$. Optical methods (fisheye photos, below-canopy PAR) give projected LAI of about 3 for SMEAR II, data not available for SMEAR I. At Zotino measurements of LAI had been performed earlier than the data set was obtained and differ from 1.3 to 3.5 $m^2$ $m^{-2}$. Therefore, in this study we prefer to rely on PAR as a robust measured parameter, while the information about the fraction of absorbed radiation, aPAR, dependent on LAI and canopy parameters, is contained in LUE = GPP/PAR, defined similarly to Cheng et al. (Using satellite-derived optical thickness to assess the inuence of clouds on terrestrial carbon uptake, JGR: Biogeosciences, 121, 1747-1761, 2016). Based on the fact that PAR dependences are rather universal, this approach allowed us to draw some general conclusions regarding GPP maximum in Section 3.2.2.

6. *Could the authors explain better the overestimation of the cloud-biased data? (line 15 page 9)*

   It follows from Fig. 5, that for clean atmosphere (low CS) and under clear sky conditions, the diffuse fraction following from the results of the clear sky modelling is small, on the order of 10%. However, many measured points, selected using simplified criterion of clear sky which includes cloud-biased points, have higher diffuse fraction values. This means that on average they result in a higher diffuse fraction 12-17% (Table 3) as compared to ~10% predicted for Rayleigh scattering conditions by the model.

7. *I believe their criteria is robust to distinguish between aerosol effects and thin clouds (line 35 and p[age 10). However, haze can be very difcult to distinguish. Could the authors comment on this point?*

If the reviewer means plumes from forest fires under haze, the clear sky model fails to give right predictions for these periods. Therefore they were excluded from consideration in the aerosol-radiation part of the study (data set from Fonovaya, 2016), which we mentioned in the manuscript.

8. *Figure 8 summarizes and it is in my opinion the highlight of the paper. However, all the data is gone and only the estimated dependences are given. Why? I understand that the data can be very scattered but I think it can be interesting for the reader to see by him/herself these maximum behaviour. The behaviour reminds me the one reported by Min and Wang (Geophysical Research Letters doi:10.1029/2007GL032398, 2008, see Figure 1). Since they dont have a discussion section, I think as a reader I will appreciate a more elaborate discussion.*

We added the figure (Fig. 9) and discussion in the manuscript (p. 14, l. 26-30). Our data sets look similar to those reported by Alton et al. (A sensitivity analysis of the land-surface scheme JULES conducted for three for- 5 est biomes: Biophysical parameters, model processes, and meteorological driving data, Global Biogeochemical Cycles, 20, 2007) and Alton (Reduced carbon sequestration in terrestrial ecosystems under overcast skies compared to clear skies, Agricultural and Forest Meteorology, 148, 1641 - 1653, 2008). The increase in GPP reported for SMEAR II is also similar to Alton (2008), but for mixed forests we obtained increase up to 30% as compared to moderate 10% increase for broadleaf forests reported by Alton (2008). Note that he used parametrization for the diffuse fraction of global radiation while we had measurements of diffuse radiation at four sites out of five.

We thank again the referee for the useful suggestions. We hope that the manuscript is now suitable for publication in ACP.